# Systematic allelic analysis defines the interplay of key pathways in X chromosome inactivation

Tatyana B. Nesterova [1,6], Guifeng Wei [1,6], Heather Coker[1,6], Greta Pintacuda[1,3,6], Joseph S. Bowness [1], Tianyi Zhang[1], Mafalda Almeida [1], Bianca Bloechl[1], Benoit Moindrot [1,4], Emma J. Carter [1], Ines Alvarez Rodrigo [1,5], Qi Pan[1], Ying Bi [2], Chun-Xiao Song [2] & Neil Brockdorff[1]

Xist RNA, the master regulator of X chromosome inactivation, acts in cis to induce chromosome-wide silencing. Whilst recent studies have defined candidate silencing factors, their relative contribution to repressing different genes, and their relationship with one another is poorly understood. Here we describe a systematic analysis of Xist-mediated allelic silencing in mouse embryonic stem cell-based models. Using a machine learning approach we identify distance to the *Xist* locus and prior gene expression levels as key determinants of silencing efficiency. We go on to show that Spen, recruited through the Xist A-repeat, plays a central role, being critical for silencing of all except a subset of weakly expressed genes. Polycomb, recruited through the Xist B/C-repeat, also plays a key role, favouring silencing of genes with pre-existing H3K27me3 chromatin. LBR and the Rbm15/m6A-methyltransferase complex make only minor contributions to gene silencing. Together our results provide a comprehensive model for Xist-mediated chromosome silencing.

[1] Developmental Epigenetics, Department of Biochemistry, University of Oxford, South Parks Road, Oxford OX1 3QU, UK. [2] Ludwig Institute for Cancer Research, Target Discovery Institute, Nuffield Department of Medicine, University of Oxford, Oxford OX3 7FZ, UK. [3] Present address: Department of Stem Cell and Regenerative Biology, Harvard University, 7 Divinity Avenue, Cambridge, MA 02138, USA. [4] Present address: I2BC Paris-Sud University, Gif-Sur-Yvette, France. [5] Present address: Dunn School of Pathology, University of Oxford, South Parks Road, Oxford, OX1 3RE, UK. [6] These authors contributed equally: Tatyana B. Nesterova, Guifeng Wei, Heather Coker, Greta Pintacuda. Correspondence and requests for materials should be addressed to N.B. (email: neil.brockdorff@bioch.ox.ac.uk)

X chromosome inactivation is a developmentally regulated process that evolved in mammals to equalise the levels of X-linked gene expression in XX females relative to XY males[1]. X inactivation is established by the X inactive specific transcript (Xist), a 17 kb non-coding RNA, which accumulates *in cis* across the entire inactive chromosome elect[2–5]. Xist RNA recruits several factors that collectively modify chromatin/chromosome structure to silence transcription[6]. Xist transgenes can recapitulate Xist function in the context of an autosome[5,7], albeit with somewhat reduced efficiency[8–10].

Xist-mediated chromosome silencing occurs in a stepwise manner with specific chromatin/chromosome modifications occurring concurrent with the onset of Xist RNA expression, and others at later timepoints[11]. Several silencing factors have been identified, including in recent studies, RNA-binding proteins (RBPs) that interact directly with specific elements within Xist RNA. The RBP Spen has been identified as an important factor for establishment of Xist-mediated silencing[12–15], functioning by recruiting the NCoR-HDAC3 complex to induce chromosome-wide histone deacetylation[15,16]. Spen binds to the Xist A-repeat, a short tandem repeat region at the 5′ end of the transcript[12,17], that is a key element required for Xist-mediated silencing[18]. Two other factors whose recruitment is linked to the A-repeat are Wtap, a regulatory subunit of the N6-Methyladenosine (m6A) methyltransferase complex[13,14], and Rbm15, an RBP related to Spen[13,19]. Wtap and Rbm15 have moreover been linked with one another, with the latter directly recruiting the m6A-methyltransferase complex via binding to Xist A-repeat[19]. Accordingly, several studies have identified sites of m6A deposition in Xist RNA[19–22]. Both Rbm15 and the m6A-methyltransferase (m6A-MTase) complex have been implicated in Xist-mediated silencing[13,14,19].

A third recently identified silencing factor is the Lamin B receptor (LBR), mutation of which has been reported to strongly abrogate Xist-mediated silencing[15,23]. LBR binds to several sites across Xist RNA, as determined by CLIP-seq, with the most prominent site, referred to as LBS, being located around 0.2 kb downstream of the A-repeat. Deletion of LBS results in abrogated silencing, which is complemented by tethering LBR to the mutant RNA[23]. However, *Lbr* mutations in mouse do not show a female-specific phenotype[24,25], suggesting that the role of the protein in X inactivation requires further investigation.

In addition to the aforementioned factors, Xist RNA recruits the Polycomb repressive complexes PRC1 and PRC2 that catalyse the histone modifications H2AK119ub and H3K27me3, respectively[26–28]. Polycomb recruitment is initiated by the RBP hnRNPK which binds to a different Xist RNA region, the B/C-repeat (herein PID region)[10]. hnRNPK interacts with the Pcgf3/5-PRC1 complex which initiates a Polycomb cascade resulting in recruitment of other PRC1 complexes and PRC2[29]. Polycomb is important for chromosome-wide silencing by Xist RNA[29], although the relative contribution of PRC1 and PRC2 has not been determined.

Although key silencing factors and their binding sites have been identified, their relationship with one another and the relative contribution they make to repressing individual genes or gene subsets is not well understood. A key limitation is that each factor/pathway has been analysed in different systems, for example XY mouse embryonic stem cells (mESCs), XX mESCs, trophectoderm tissue of XX embryos or XX somatic cells, and/or at different timepoints. Additionally, several different approaches have been used to assay silencing, including nascent RNA-FISH/single molecule FISH of selected genes, chromosome-wide analysis, either with or without an interspecific background to determine allelic ratios, and indirect assays, for example escape from cell death following induction of Xist RNA expression on the single X chromosome in XY mESCs.

To define the contribution of the different pathways in X chromosome inactivation, we present here a systematic chromosome-wide analysis of Xist-mediated silencing in two complementary interspecific mouse ES cell models, carrying either a transgenic inducible Xist on an autosome (chromosome 3) or inducible endogenous Xist in XX mESCs. Our results demonstrate that establishment of Xist-mediated silencing is largely attributable to synergistic actions of Spen/A-repeat and Polycomb/B-repeat.

## Results

**The dynamics of Xist-mediated silencing in mESC models.** To define the contribution of different pathways in Xist-mediated silencing we analysed two complementary models, a previously described XY interspecific M.m. domesticus (129S) × M.m. castaneus (Cast) mESC line with a multicopy tetracycline inducible Xist transgene randomly integrated on chromosome 3 (refs. [10,30]), referred to herein as iXist-Chr3, and a novel 129S × Cast XX ES cell line (iXist-ChrX) with a single endogenous *Xist* allele driven by a tetracycline inducible promoter (Fig. 1a). Thus, an expression construct encoding the TetOn transactivator was introduced into the TIGRE locus[31], enabling efficient and synchronous induction of Xist RNA (129S allele) in ~95% of cells (Supplementary Fig. 1a, b). Subsequent allelic sequencing analysis (see below) revealed that a recombination event between 129S and Cast X chromosomes occurred in the process of modifying the Xist promoter such that informative SNPs are present only for the 103 Mb proximal to *Xist* (Fig. 1a).

Xist-mediated gene silencing was assayed using ChrRNA-seq, a sensitive method that utilises abundant intronic, in addition to exonic SNPs, to enable accurate determination of allelic expression of the majority of genes[32]. Thus, we analysed iXist-Chr3 mESCs 1 day and 3 days after Xist RNA induction, the latter in differentiating conditions. Subsequent timepoints were not analysed as selection against autosomal monosomy progressively depletes the proportion of cells with detectable Xist RNA expression. The location of the Xist transgene (125.8097Mb–125.8145Mb, mm10, Cast allele) was determined using chimeric Xist-Chr3 paired-end sequence tags from ChIP-seq analysis (see below). Silencing levels were determined for genes which had at least 10 SNP informative reads ($n$~500). As shown in Fig. 1b, c, at 1 day we observed silencing of most genes chromosome-wide, albeit to a reduced degree at the distal end of the chromosome. Silencing was further reinforced at 3 days (Fig. 1b, c).

iXist-ChrX mESCs were analysed after 1 day of Xist RNA induction, and then at a later timepoint, 6 days, under differentiating conditions. Silencing levels were again determined for genes which had at least 10 SNP informative reads ($n$~250). At 1 day we observed significant silencing of most genes using both allelic analysis for *Xist* proximal genes (Fig. 1d, e), and using non-allelic analysis for the entire ChrX (Supplementary Fig. 1c, d). Overall silencing efficiency was greater than that observed with iXist-Chr3 (Fig. 1f), consistent with previous reports that Xist RNA functions more efficiently on ChrX[8–10]. Silencing in iXist-ChrX cells was more robust after 6 days of Xist RNA induction (Fig. 1d, e), although interestingly, for many genes silencing is incomplete relative to XX somatic cells (Supplementary Fig. 1e).

We went on to explore if specific gene/chromatin characteristics (2D/3D distance relative to *Xist*, mESC gene expression level, and chromatin features of genes centred on promoters), correlate with silencing efficiency in iXist-Chr3 and iXist-ChrX cells 1 day after Xist RNA induction. In the case of 3D distance from *Xist*, recent work has shown that Xist RNA spreading on the X chromosome is initially determined by topological proximity to the *Xist* locus[33]. We were able to demonstrate a similar

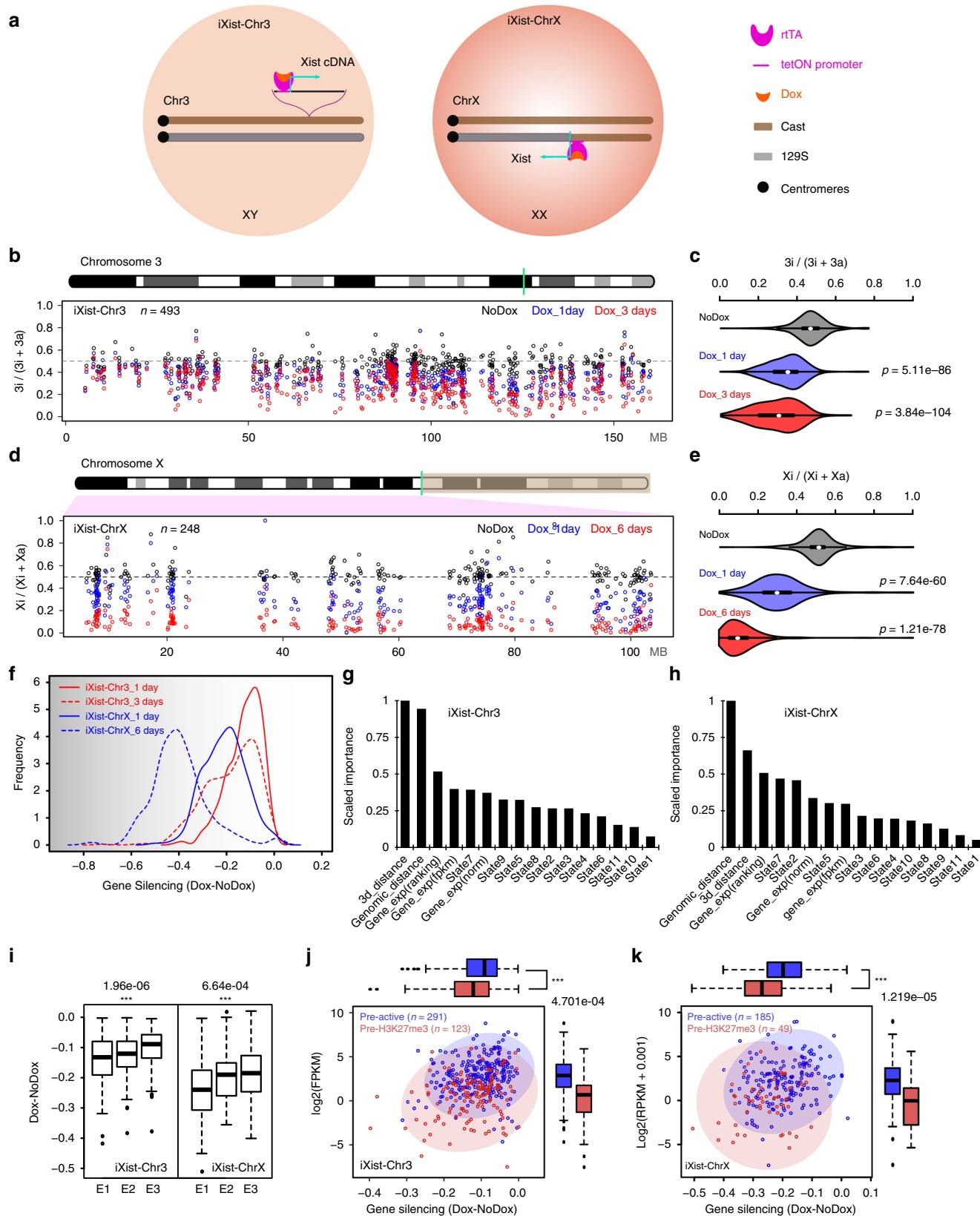

correlation for iXist-Chr3 cells using, as a surrogate marker, Polycomb-mediated histone modifications, H2AK119ub and H3K27me3, determined using allelic calibrated ChIP-seq (Supplementary Fig. 1f). Using a published cohesin HiChIP dataset[34], we found a close correlation ($r = 0.76$–$0.77$) between Xist-dependent gain of Polycomb-mediated histone modifications

and topological proximity to the Xist transgene (Supplementary Fig. 1f, g). To classify gene promoter chromatin states, we employed ChromHMM[35] to integrate multiple ENCODE ChIP-seq datasets defining chromatin states in mESC. Genes were further classified according to the chromatin state surrounding the TSS region (see Methods).

**Fig. 1** mESC models for analysis of Xist-mediated silencing. **a** Schematic summarising engineering of iXist-Chr3 and iXist-ChrX mESCs. **b** Allelic silencing across Chr3 in uninduced (NoDox), 1 day and 3 days induced (Dox) iXist-Chr3 cells. Mean value of allelic ratio for each gene with an informative SNP was calculated from biological replicates as detailed in Supplementary Table 4. Green line on Chr3 ideogram indicates location of Xist transgene. **c** Violin plot summarising distribution of ratio values in **b**. *p* values were calculated by one-sided Wilcoxon rank-sum test. **d** Allelic silencing on ChrX in uninduced (NoDox), 1 day and 6 days induced (Dox) iXist-ChrX cells. Mean value of allelic ratio for each gene with an informative SNP was calculated from biological replicates as detailed in Supplementary Table 4. Green line on ChrX ideogram indicates location of Xist locus. **e** Violin plot summarising data in **d**. *p* values were calculated by one-sided Wilcoxon rank-sum test. **f** Comparison of gene silencing in iXist-Chr3 and iXist-ChrX cell lines. **g**, **h** Importance of parameters for predicting silencing efficiency in iXist-Chr3 (**g**) and iXist-ChrX (**h**) cells using machine learning. AUC (area under the curve) is 0.70 and 0.71 for iXist-Chr3 and iXist-ChrX, respectively. **i** Allelic silencing for quantiles E1–E3 based on mESC gene expression level in iXist-Chr3 and iXist-ChrX mESCs 1 day after Xist RNA induction, where E1 is lower expressed gene group and E3 is the highest, with significance indicated using one-way ANOVA test. **j**, **k** Scatterplots and boxplots illustrating more efficient silencing of genes that have pre-existing mESC H3K27me3 (pre-H3K27me3) chromatin/low expression level in iXist-Chr3 cells (**j**) and iXist-ChrX cells (**k**). *p* values calculated using two-sided Wilcoxon rank-sum test. For all the boxplots (**i**–**k**) the lower and upper edge of the box represent the first and third quartile, respectively. The horizontal line inside the box indicates the median. Whiskers identify the farthest data points within 1.5× the interquartile range (IQR)

To rank the contribution of the aforementioned parameters to silencing efficiency after 1 day of Xist RNA induction, we applied a machine learning approach to train a classifier to discriminate high or low silencing efficiency based on combinatory features for both iXist-Chr3 and iXist-ChrX lines (see Methods), and then scored the importance of each feature (Fig. 1g, h). The most informative feature for Xist-mediated chromosome silencing is 2D/3D proximity to *Xist*. The correlation for 3D proximity in relation to silencing efficiency is shown for iXist-Chr3 ($r = 0.41$) and iXist-ChrX ($r = 0.41$) (Supplementary Fig. 1g, h). mESC gene expression levels also ranked as being of relatively high importance (Fig. 1g, h). Further analysis demonstrated that this reflects more efficient silencing of weakly expressed genes (Fig. 1i). Accordingly, promoter proximal chromatin states that correlated with silencing efficiency both in iXist-Chr3 and iXist-ChrX were state 7 (pre-active, negative correlation) and state 5 (pre-H3K27me3, positive correlation). Further analysis shows that this reflects more efficient silencing of pre-H3K27me3 associated genes (typically weakly expressed), and less efficient silencing of pre-active genes (Fig. 1j, k, Supplementary Fig. 1i, j).

**A key role for Spen and the Xist A-repeat in gene silencing**. We generated a null allele of *Spen* by CRISPR/Cas9 mutagenesis essentially as described previously[12], initially in iXist-Chr3 mESCs (see Methods and Supplementary Fig. 2a–c). iXist-Chr3 *Spen* null cells were viable and morphologically indistinguishable from unmodified controls. Robust Xist RNA domains were formed in the majority of cells following transgene induction (Supplementary Fig. 2d), although levels of Xist RNA were reduced relative to wild-type (WT) iXist-Chr3 mESCs (Supplementary Fig. 2e). We went on to analyse allelic silencing in iXist-Chr3ΔSpen mESCs 1 day after Xist RNA induction. Highly reproducible results, obtained using four independent *Spen* null mESC lines (see Methods), were averaged. As shown in Fig. 2a, b, silencing of genes across chromosome 3 was strongly abrogated in the iXist-Chr3 *Spen* null cells, in most cases to the extent that expression levels were indistinguishable from uninduced control cells. To quantify our findings we defined thresholds to assign silencing degree as high, low and weak/none (Methods). On this basis a subset of chromosome 3 genes were silenced (24.8%; 127/512). In most cases the degree of silencing was classified as low (119/127). Further analysis revealed that genes subject to silencing in *Spen* null cells are generally weakly expressed in mESCs (Fig. 2c, left panel).

We also deleted *Spen* in iXist-ChrX mESCs. Viability and ES cell morphology were again unaffected, as was the frequency of cells inducing Xist RNA expression 1 day after addition of doxycycline (Supplementary Fig. 2a–d). The levels of induced Xist RNA were relatively low, similar to *Spen* null iXist-Chr3 cells

(Supplementary Fig. 2e), and additionally, Xist RNA domains were in some instances relatively weak and diffuse compared with WT iXist-ChrX cells (Supplementary Fig. 2d). Allelic chromatin RNA-seq performed 1 day after Xist RNA induction (three independent *Spen* null cell lines) demonstrated a dramatic reduction in Xist-mediated silencing, similar to *Spen* null iXist-Chr3 cells (Fig. 2d, e). Also, as noted for *Spen* null iXist-Chr3 cells, a subset of genes (40/261) exhibited a low degree of silencing. In most cases these genes are weakly expressed in mESCs (Fig. 2c, right panel).

We went on to compare the effect of *Spen* deletion in iXist-ChrX cells with deletion of the Xist A-repeat, the critical silencing element linked to Spen recruitment[12,14]. A deletion of the A-repeat, (XistΔA), was engineered in iXist-ChrX cells using CRISPR/Cas9-mediated mutagenesis (Supplementary Fig. 2f, g). This deletion is somewhat shorter than previously published A-repeat deletions, which in most cases additionally remove 185nt of sequence 3′ to the A-repeat element. This point is significant in relation to other experiments described below. Xist RNA-FISH analysis demonstrated formation of Xist domains at comparable frequency as in WT iXist-ChrX cells, although similar to *Spen* null cells, levels of Xist RNA were significantly reduced (Supplementary Fig. 2e). Allelic ChrRNA-seq in XistΔA mESCs after 1 day of Xist RNA induction demonstrated strongly abrogated silencing for most genes, similar to the *Spen* null iXist-ChrX cells (Fig. 2e, f). Induction of Xist RNA for 6 days in differentiating conditions gave similar results (Supplementary Fig. 2h, i). In both cases a subset of genes (12.3%; 32/261 at day 1) undergo some degree of silencing, and as noted for *Spen* null iXist-Chr3 and iXist-ChrX cell lines, these correlate with genes that are expressed weakly in mESCs (Fig. 2c, right panel). Most genes silenced in *Spen* null iXist-ChrX cells are also silenced in XistΔA mESCs ($p = 0.012$; hypergeometric test, Supplementary Fig. 2j). Similarly, the subset of genes silenced in A-repeat mutant iXist-ChrX cells after 1 day or 6 days of Xist RNA induction also correlate (Supplementary Fig. 2k). Taken together, our findings demonstrate that Xist A-repeat and the A-repeat bound factor Spen play a major role in silencing at all except a small subset of weakly expressed genes.

**Minor role of m6A-MTase and LBR in Xist-mediated silencing**. Rbm15, a homologue of Spen, has been implicated in Xist-mediated silencing, and has been proposed to function by recruiting the m6A-MTase complex[13,14,19]. To determine the contribution of Rbm15/m6A-MTase complex in Xist-mediated silencing, we generated knockouts of the *Rbm15*, *Mettl3* and *Wtap* genes in iXist-Chr3 cells, using CRISPR/Cas9-mediated mutagenesis (Supplementary Fig. 3a). None of the mutations had detectable effects on Xist RNA domains (Supplementary Fig. 3b)

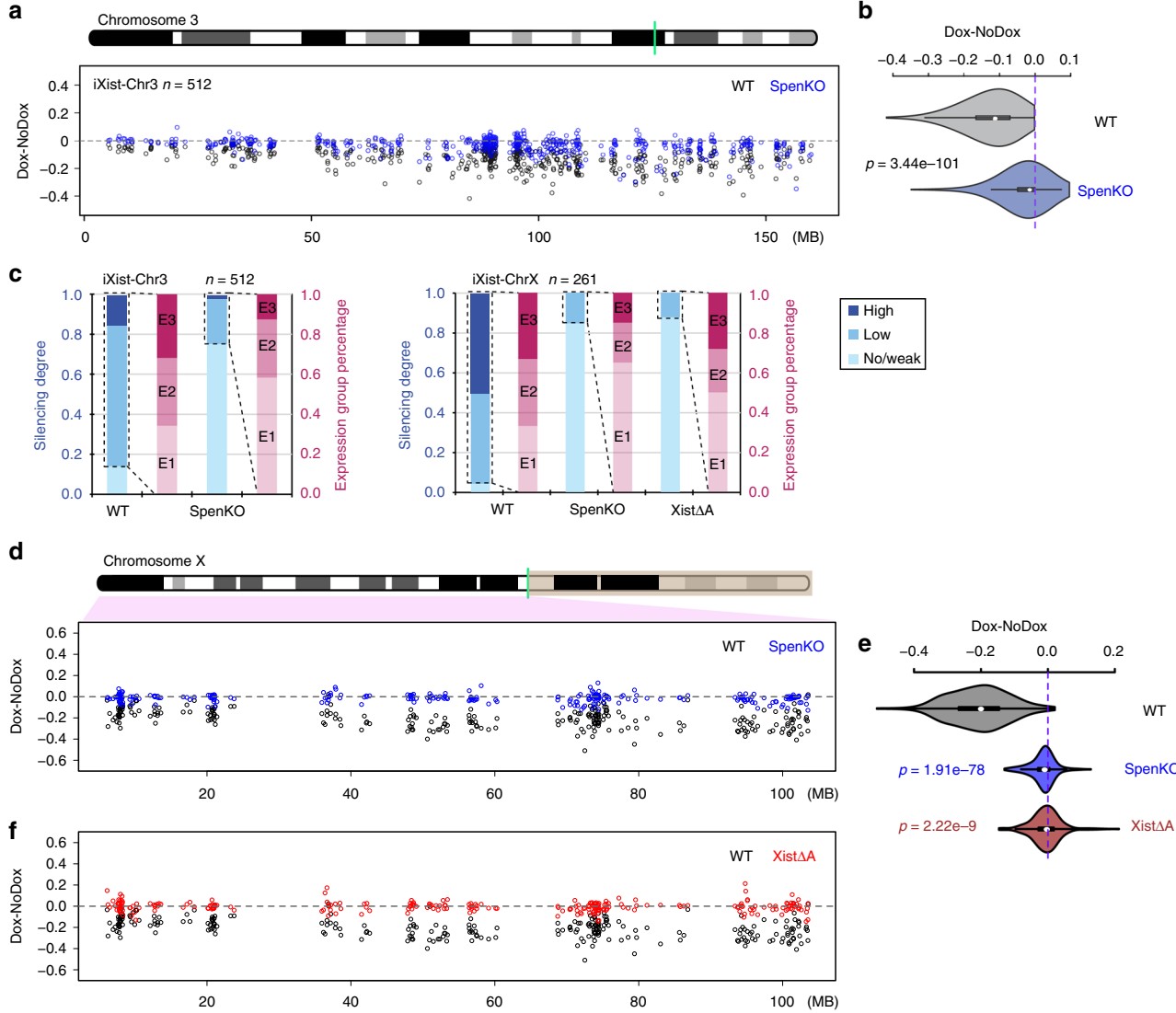

**Fig. 2** Effect on Xist-mediated silencing after deletion of Spen/A-repeat. **a** Allelic silencing across Chr3 after 1 day of Xist RNA induction in WT and *Spen* null (KO) iXist-Chr3 cells. Mean value of difference of allelic ratio for each gene with an informative SNP was calculated from biological replicates as detailed in Supplementary Table 4. Green line on Chr3 ideogram indicates location of Xist transgene. **b** Violin plot summarising data in **a**, *p* values were calculated using one-sided Wilcoxon rank-sum test. **c** Distribution of genes by silencing degree in relation to mESC expression levels (three equal sized quantiles, E1 < E2 < E3) in WT and *Spen* null (KO) iXist-Chr3 cells (left panel) and WT, XistΔA and *Spen* null iXist-ChrX cells (right panel). **d** Allelic silencing across ChrX after 1 day of Xist RNA induction in WT and *Spen* null (KO) iXist-ChrX cells. Mean value of difference of allelic ratio for each gene with an informative SNP was calculated from biological replicates as detailed in Supplementary Table 4. Green line on ChrX ideogram indicates location of the *Xist* locus. **e** Violin plot summarising data in **d** and **f** . *P* values were calculated using one-sided Wilcoxon rank-sum test. **f** Allelic silencing across ChrX after 1 day of Xist RNA induction in WT and XistΔA iXist-ChrX cells. Mean value of difference of allelic ratio for each gene with an informative SNP was calculated from biological replicates as detailed in Supplementary Table 4. Green line on ChrX ideogram indicates location of the *Xist* locus

or Xist RNA levels (Supplementary Fig. 2e). As above, we assessed Xist-mediated silencing using allelic ChrRNA-seq after 1 day of Xist RNA induction. Surprisingly, none of the three mutations abrogated gene silencing by Xist RNA, and in the case of *Rbm15* null mESCs, silencing efficiency was if anything enhanced (Fig. 3a, b, Supplementary Fig. 3c). To investigate how the mutations affect m6A genome-wide we performed m6A quantitation using HPLC MS/MS and m6A-seq (as described in ref. [36]), in the *Mettl3* and *Wtap* knockout lines. Notably, we found that whilst overall m6A levels are reduced (Supplementary Fig. 3d), residual m6A is detectable at some peaks, including the major peak in Xist RNA, which maps immediately downstream of the A-repeat (Supplementary Figs. 3e and 2f). This observation likely reflects non-essentiality in the case of the *Wtap* knockout. For the

*Mettl3* knockout, residual m6A may indicate partial redundancy with the Mettl14 subunit, or alternatively, that the allele we analysed is hypomorphic due to the presence of additional AUG sites (see Supplementary Fig. 3a). Despite repeated efforts we were unable to generate ES cells entirely lacking a functional Rbm15/m6A-MTase complex (knockout of *Rbm15* together with its homologue *Rbm15b*, deletion of the entire *Mettl3* locus, and knockout of the *KIAA1429/virilizer* gene, encoding another core subunit of the complex), suggesting that residual activity of the complex may be essential for ES cell viability. This is likely a consequence of the key role m6A plays in several steps of mRNA biogenesis, and is in keeping with a previous report[19].

We went on to derive and validate *Rbm15* and *Wtap* null iXist-ChrX cell lines (Supplementary Fig. 3a, b), and performed allelic

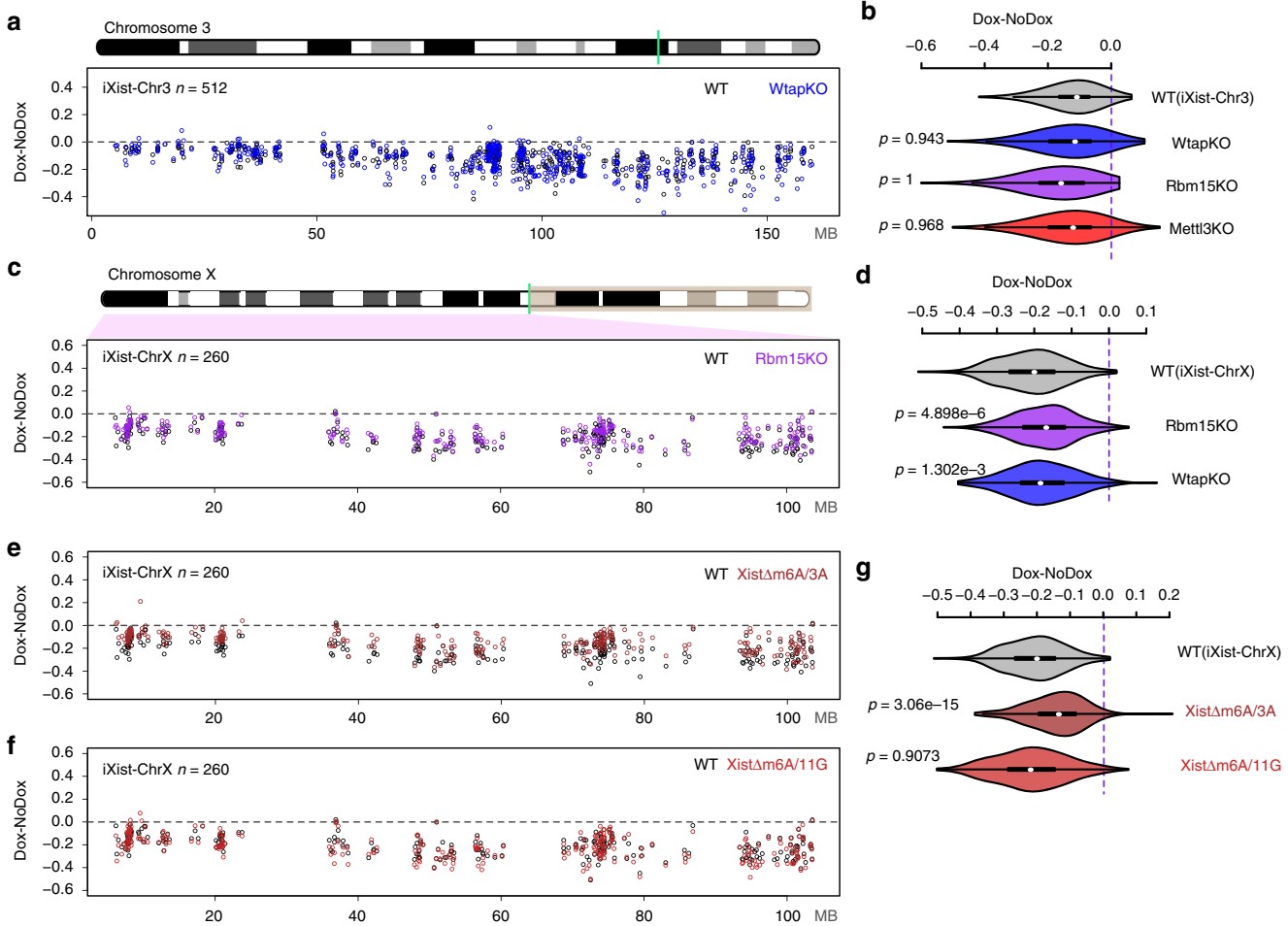

**Fig. 3** The role of Rbm15/m6A-MTase complex in Xist-mediated silencing. **a** Allelic silencing across Chr3 after 1 day of Xist RNA induction in WT and *Wtap* null (KO) iXist-Chr3 cells. Mean value of difference of allelic ratio for each gene with an informative SNP was calculated from biological replicates as detailed in Supplementary Table 4. Green line on Chr3 ideogram indicates location of the Xist transgene. **b** Violin plot summarising data for *Wtap* null in **a**, and additionally, data for *Mettl3* mutant and *Rbm15* null shown in Supplementary Fig. 3c. *P* values were calculated using one-sided Wilcoxon rank-sum test. **c** Allelic silencing across ChrX after 1 day of Xist RNA induction in WT and *Rbm15* null (KO) iXist-ChrX cells. Mean value of difference of allelic ratio for each gene with an informative SNP was calculated from biological replicates as detailed in Supplementary Table 4. Green line on ChrX ideogram indicates location of the *Xist* locus. **d** Violin plot summarising data in **c**. *P* values were calculated using one-sided Wilcoxon rank-sum test. **e**, **f** Allelic silencing across ChrX after 1 day of Xist RNA induction in WT and cell lines with deletions within the major m6A peak region, XistΔm6A/3A (**e**) and XistΔm6A/11 G (**f**) in iXist-ChrX cells. Mean value of difference of allelic ratio for each gene with an informative SNP was calculated from biological replicates as detailed in Supplementary Table 4. **g** Violin plot summarising data in **e**, **f**. *P* values were calculated using one-sided Wilcoxon rank-sum test

ChrRNA-seq 1 day after Xist RNA induction. In contrast to iXist-Chr3 cells we observed a small but reproducible reduction in silencing across the chromosome, both for *Rbm15* and *Wtap* null cell lines (Fig. 3c, d, Supplementary Fig. 3c). We did not observe differential effects based on classification of ChrX genes by position, expression level, pre-existing chromatin state, or silencing efficiency.

Given the different effects on Xist-mediated silencing reported above and in prior studies[13,19], together with the fact that it was not possible to assess complete loss of function of the Rbm15/m6A-MTase complex, we went on to determine the effect of deleting m6A sites in the 5′ region of Xist RNA in the iXist-ChrX cell line. Thus, we generated and validated two cell lines XistΔm6A/3A and XistΔm6A/11G, with overlapping short deletions of this region (Supplementary Figs. 2e–g and 3b). We then assayed Xist-mediated silencing 1 day after induction using allelic ChrRNA-seq. The results are summarised in Fig. 3e–g. The most proximal deletion (XistΔm6A/3A) resulted in a small deficit in Xist-mediated silencing across the length of the chromosome, consistent with

the effects seen in *Rbm15* and *Wtap* null iXist-ChrX cells. In contrast, the distal deletion (XistΔm6A/11G) had no detectable effect on chromosome-wide silencing. The latter observation may indicate a hierarchical relationship of different m6A target sites, or alternatively that the proximal deletion is closer to the A-repeat and disrupts normal function of this element. Taken together our results indicate that the Rbm15/m6A-MTase complex plays only a minor role in Xist-mediated chromosome silencing (see also Discussion).

To investigate the contribution of LBR to Xist-mediated silencing we generated a null allele of the *Lbr* gene by deleting the entire locus using CRISPR/Cas9-mediated mutagenesis, initially in iXist-Chr3 mESCs (Supplementary Fig. 4a–c). The deletion had no effect on Xist transgene expression (Supplementary Figs. 2e and 4d). Strikingly, allelic ChrRNA-seq 1 day after Xist RNA induction revealed no detectable effects on Xist-mediated silencing (Fig. 4a, b). To test if LBR may be important at a later stage, we repeated the ChrRNA-seq experiment 3 days after Xist RNA induction in differentiating conditions, but again saw no effect on chromosome-wide silencing (Supplementary Fig. 4e, f).

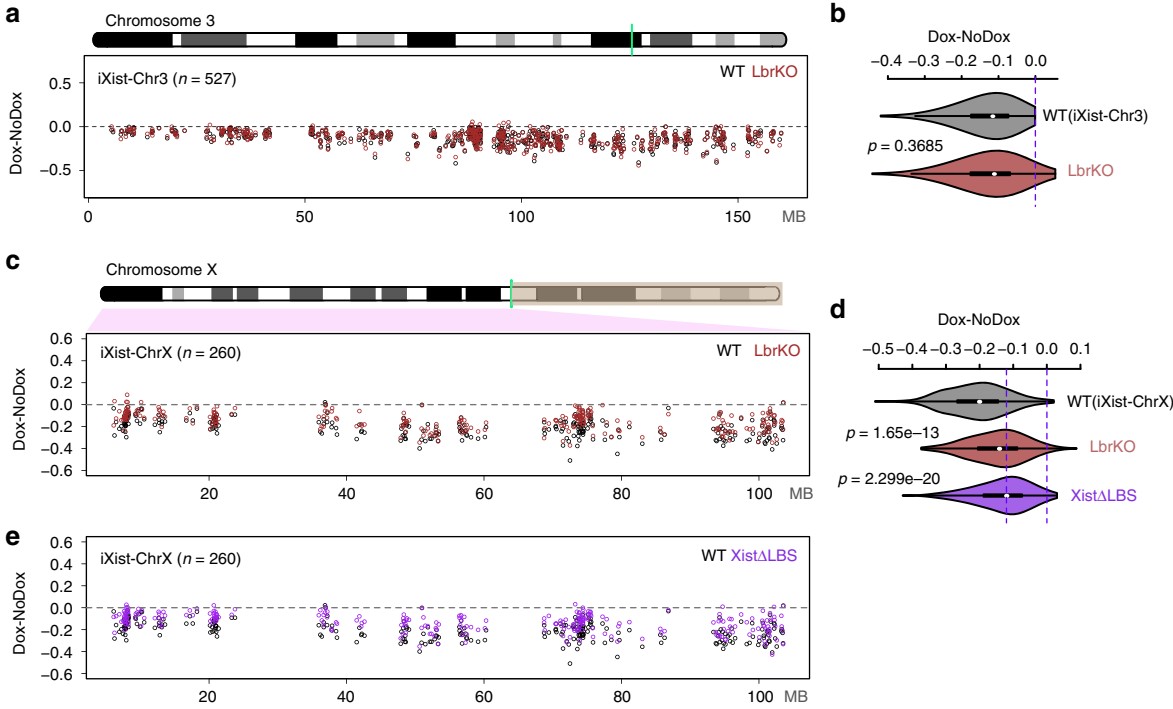

**Fig. 4** The role of LBR in Xist-mediated silencing. **a** Allelic silencing across Chr3 after 1 day of Xist RNA induction in WT and *Lbr* null (KO) iXist-Chr3 cells. Mean value of difference of allelic ratio for each gene with an informative SNP was calculated from biological replicates as detailed in Supplementary Table 4. Green line on Chr3 ideogram indicates location of the Xist transgene. **b** Violin plot summarising data in **a**. P values were calculated using one-sided Wilcoxon rank-sum test. **c** Allelic silencing across ChrX after 1 day of Xist RNA induction in WT and *Lbr* null (KO) iXist-ChrX cells. Mean value of difference of allelic ratio for each gene with an informative SNP was calculated from biological replicates as detailed in Supplementary Table 4. Green line on ChrX ideogram indicates location of the *Xist* locus. **d** Violin plot summarising data in **c** and **e**. P values were calculated using one-sided Wilcoxon rank-sum test. **e** Allelic silencing across ChrX after 1 day of Xist RNA induction in WT and XistΔLBS iXist-ChrX cells. Mean value of difference of allelic ratio for each gene with an informative SNP was calculated from biological replicates as detailed in Supplementary Table 4

We went on to investigate LBR function in Xist-mediated silencing in the context of ChrX. Thus, we generated and validated *Lbr* null iXist-ChrX cell lines (Supplementary Fig. 4a±d).

Assaying Xist-mediated silencing by ChrRNA-seq revealed a small reduction in silencing across the X chromosome (Fig. 4c, d). No differential effects were observed in relation to gene location, expression level, or silencing efficiency, although there was a minor preference towards silencing of genes with a pre-existing H3K27me3 chromatin environment (see below).

LBR has been proposed to directly bind to Xist RNA via the LBS element located between the A- and B-repeats[23]. We therefore generated and validated an LBS deletion in iXist-ChrX cells (Supplementary Figs. 2f–g and 4g), and performed allelic ChrRNA-seq after 1 day of Xist induction (Fig. 4d, e). Xist-mediated silencing was more strongly affected than in *Lbr* null iXist-ChrX cells, although it was nevertheless a relatively weak effect. The correlation of gene silencing in XistΔLBS and *Lbr* null iXist-ChrX is significant ($r = 0.72$), but not exact (Supplementary Fig. 4h). The latter observations indicate that recruitment of LBR may not be the sole function of the LBS region, and LBS, encompassing the entire F-repeat, may also recruit another factor (s). No differential effects were observed for gene silencing in XistΔLBS in relation to gene location, expression level, pre-existing chromatin environment, or silencing efficiency.

**The role of PRC1 and PRC2 in Xist-mediated silencing.** The Polycomb complexes PRC1 and PRC2 which catalyse the histone modifications H2AK119ub and H3K27me3, respectively, have been linked to Xist-mediated silencing in several independent

studies. Recently we demonstrated that a specific PRC1 complex, Pcgf3/5-PRC1, recruited by hnRNPK bound to the Xist B/C-repeat region, initiates recruitment of both PRC1 and PRC2 complexes[10,29]. We therefore set out to make *Pcgf3/5* double-knockout (DKO) in iXist-Chr3 cells. In prior studies it was necessary to use a conditional knockout strategy to generate *Pcgf3/5* DKO cells[29]. We therefore developed a *Pcgf3* allele tagged with an Auxin-inducible degron[37], on a *Pcgf5* null background (Fig. 5a, Supplementary Fig. 5a–d). The GFP-degron-tagged Pcgf3 protein was degraded rapidly, within 2 h of addition of auxin (Supplementary Fig. 5c). Acute Pcgf3/5 loss of function led to a significant reduction in global levels of H2AK119ub (Supplementary Fig. 5e). Using calibrated allelic ChIP-seq analysis (see methods), we found that there was no gain of H2AK119ub or H3K27me3 on the Cast Chr3 after induction of Xist RNA for 1 day (Supplementary Fig. 5f), in accord with our previous analysis demonstrating that Pcgf3/5-PRC1 activity is upstream of PRC2/H3K27me3 in Xist-mediated silencing. We went on to perform allelic ChrRNA-seq 1 day after Xist RNA induction and analysed the effect on Xist-mediated silencing. Consistent with our prior analysis, silencing was significantly reduced across the length of the transgene-bearing chromosome 3 (Fig. 5b, c).

Because the *Pcgf3/5* DKO abolishes Xist-dependent H3K27me3 and H2AK119ub, it is not possible to determine the relative contribution of the two histone modifications to gene silencing. To address this we made use of a previously described iXist-Chr3 cell line in which the gene encoding Suz12, a core subunit of PRC2 required for catalytic activity, is deleted, and as a consequence, H3K27me3 is completely abolished (ref. [30] and see Supplementary Fig. 5e, f). Allelic ChrRNA-seq, performed

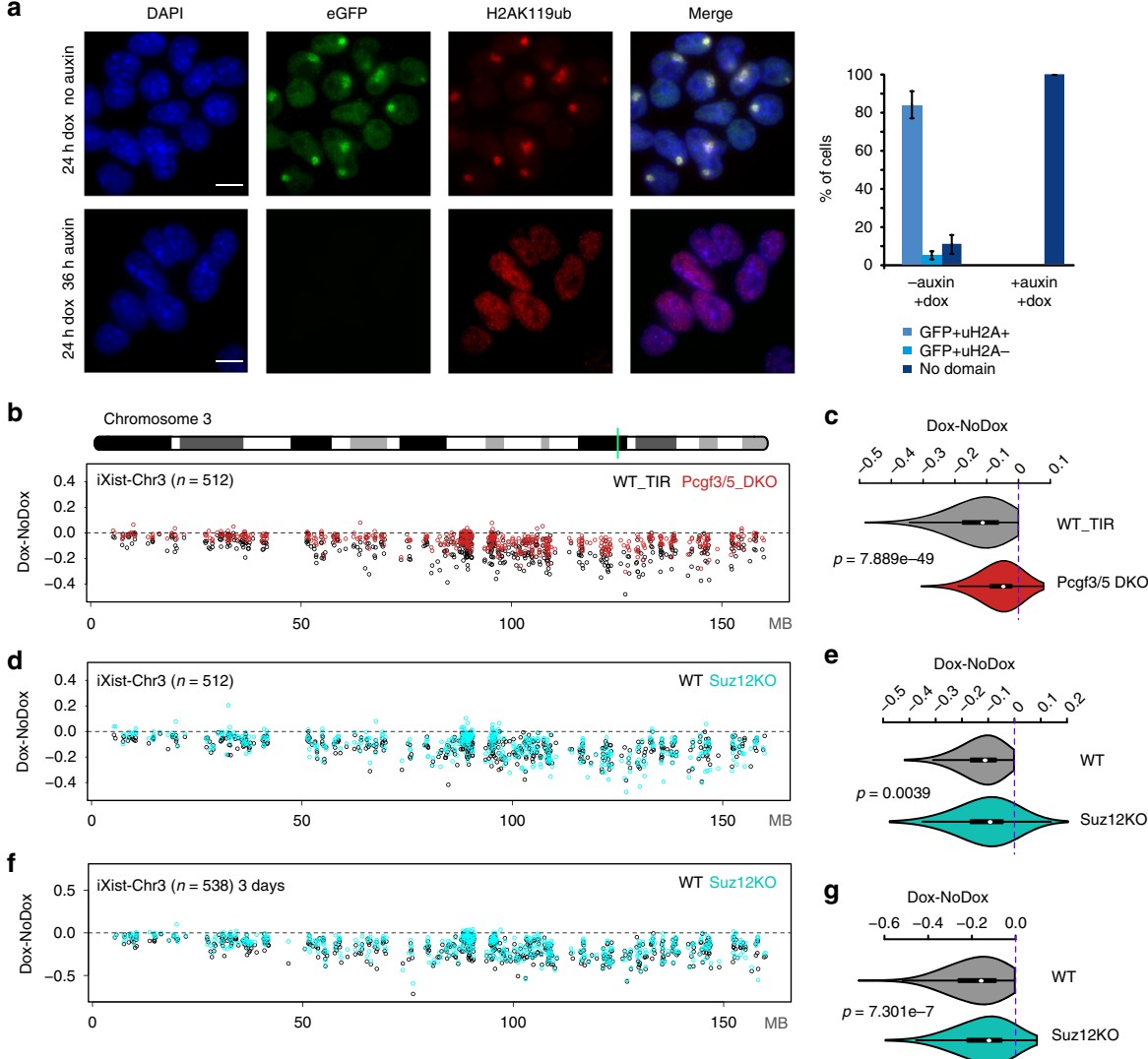

**Fig. 5** Divergent roles for PRC1 and PRC2 in Xist-mediated silencing. **a** Immunofluorescence illustrating Auxin-induced degradation of Pcgf3-GFP-AID (green) and resulting loss of Xi-H2AK119ub domains (red) as seen in iXist-Chr3 Pcgf5 null mESCs following Xist RNA induction for 1 day. Bar chart on the right shows proportion of cells with indicated phenotype in control (−auxin, +dox) and after 36 h of auxin treatment (+auxin, +dox) based on scoring >40 cells in three independent experiments. Error bars indicate standard deviation. DNA is counterstained with DAPI (blue). Scale bar is 10 μm. **b** Allelic silencing across Chr3 after 1 day of Xist RNA induction in WT and *Pcgf3/5* DKO iXist-Chr3 cells. Mean value of difference of allelic ratio for each gene with an informative SNP was calculated from biological replicates as detailed in Supplementary Table 4. Green line on Chr3 ideogram indicates location of the Xist transgene. **c** Violin plot summarising data in **b**. *P* values were calculated using one-sided Wilcoxon rank-sum test. **d** Allelic silencing across Chr3 after 1 day of Xist RNA induction in WT and *Suz12* null (KO) iXist-Chr3 cells. Mean value of difference of allelic ratio for each gene with an informative SNP was calculated from biological replicates as detailed in Supplementary Table 4. **e** Violin plot summarising data in **d**. *P* values were calculated using one-sided Wilcoxon rank-sum test. **f** Allelic silencing across Chr3 after 3 days Xist RNA induction in WT and *Suz12* null (KO) iXist-Chr3 cells. Mean value of difference of allelic ratio for each gene with an informative SNP was calculated from biological replicates as detailed in Supplementary Table 4. **g**, Violin plot summarising data in (**f**). *P* values were calculated using one-sided Wilcoxon rank-sum test

1 day after Xist induction, revealed a very small reduction in Xist-mediated silencing (Fig. 5d, e). Given prior evidence suggesting that PRC2 is important for X inactivation in vivo[26,38], we also analysed *Suz12* null cells 3 days after Xist RNA induction (under differentiating conditions). As illustrated in Fig. 5f, g and Supplementary Fig. 5g, Suz12 knockout does compromise silencing after 3 days, although only to a minor degree compared with Pcgf3/5 loss of function cells (compare Figs. 5c, g). Further analysis demonstrated that genes affected by PRC2 loss after 3 days of Xist RNA induction (63/538) are generally expressed at a relatively high level in mESCs (Supplementary Fig. 5h–j). Consistently, we found that active chromatin regions (determined by ChromHMM analysis) show a greater gain of PRC1-mediated

H2AK119ub in WT compared to Suz12 null mESCs (Supplementary Fig. 5k), suggesting that PRC2-mediated H3K27me3 plays a role in enhancing Xist-dependent gain of H2AK119ub to silence highly expressed genes. Together these experiments suggest that PRC1/H2AK119ub is the critical determinant of Polycomb function in the context of Xist-mediated silencing after 1 day of Xist RNA induction, but that PRC2/H3K27me3 plays a supporting role in maintaining/reinforcing gene repression at later timepoints.

Next we sought to assess the contribution of Polycomb in the context of Xist-mediated silencing on the X chromosome. Thus, we generated and validated a deletion encompassing the B/C-repeat in iXist-ChrX ES cells (Supplementary Figs. 2f, g and 6a).

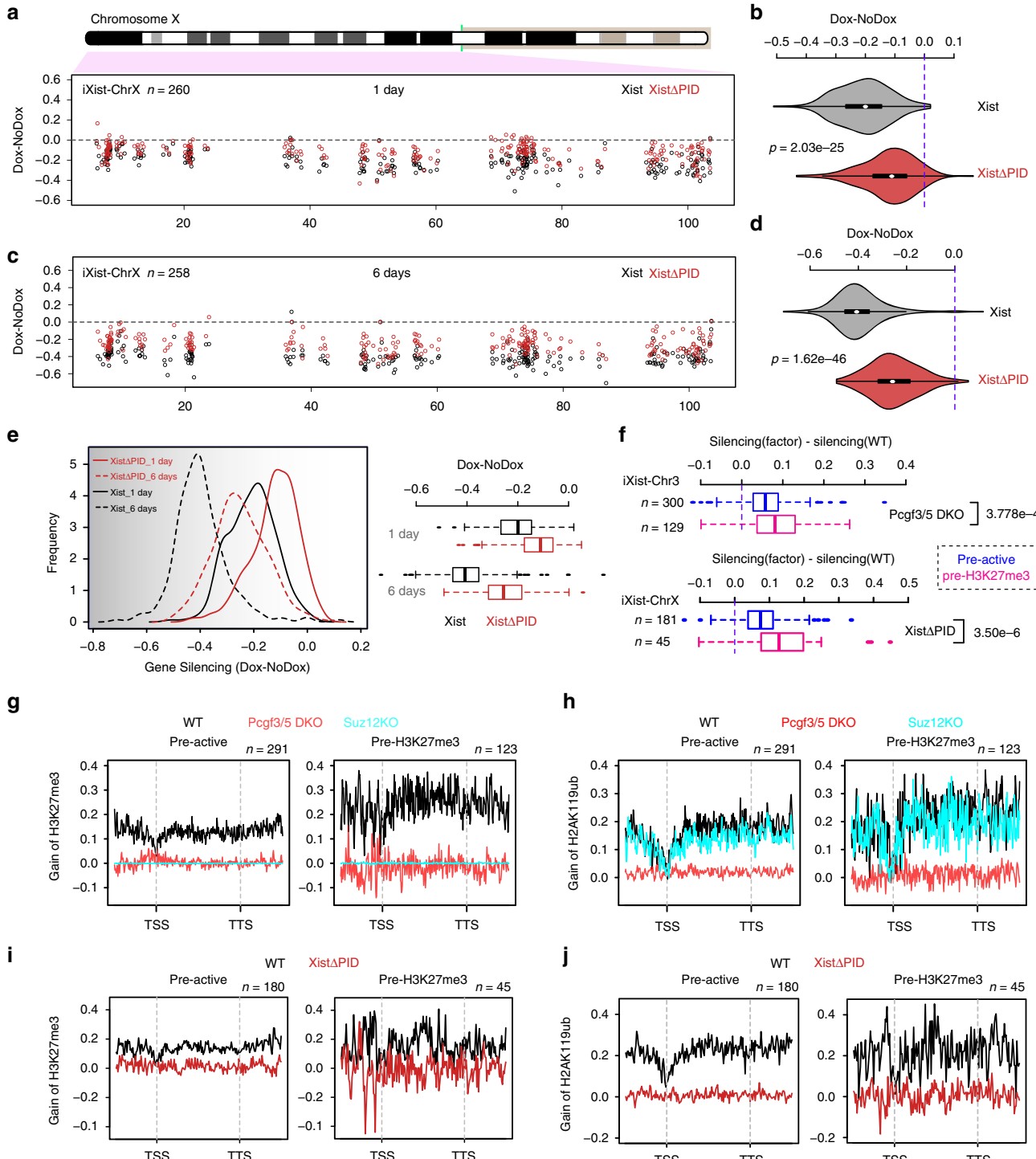

**Fig. 6** The role of Polycomb complexes in Xist-mediated silencing. **a** Allelic silencing across ChrX after 1 day of Xist RNA induction in WT and XistΔPID iXist-ChrX cells. Mean value of difference of allelic ratio for each gene with an informative SNP was calculated from biological replicates as detailed in Methods. Green line on ChrX ideogram indicates location of the *Xist* locus. **b** Violin plot summarising data in **a**. *P* values were calculated using one-sided Wilcoxon rank-sum test. **c** Allelic silencing across ChrX after 6 days of Xist RNA induction in WT and XistΔPID iXist-ChrX cells. Mean value of difference of allelic ratio for each gene with an informative SNP was calculated from biological replicates as detailed in Supplementary Table 4. **d** Violin plot summarising data in **c**. *P* values were calculated using one-sided Wilcoxon rank-sum test. **e** Comparison of chromosome silencing in WT and XistΔPID after 1 day and 6 days of Xist RNA induction (left panel). Boxplot representation of data (right panel). **f** Comparison of silencing for genes with pre-existing active (pre-active) or H3K27me3 (pre-H3K27me3) chromatin signatures over promoters for iXist-Chr3 in Pcgf3/5 DKO (top), and for iXist-ChrX in XistΔPID (bottom) after 1 day of Xist RNA induction. *P* values were calculated using two-sided Wilcoxon rank-sum test. The lower and upper edge of the box in the boxplots (**e, f**) represent the first and third quartile, respectively. The horizontal line inside the box indicates the median. Whiskers identify the farthest data points within 1.5× the interquartile range (IQR). **g, h** Metaprofile showing gain of H3K27me3 (**g**) or H2AK119ub (**h**) over genes with pre-existing active or H3K27me3 chromatin in WT, Pcgf3/5 DKO or Suz12 KO iXist-Chr3 ES cells 1 day after Xist RNA induction. **i, j** Metaprofile showing gain of H3K27me3 (**i**) or H2AK119ub (**j**) over genes with pre-existing active and H3K27me3 chromatin following 1 day of Xist RNA induction in WT and XistΔPID mESCs

This deletion, which extends into the downstream C-repeat, is equivalent to the Xist transgene deletion ΔPID that we previously showed abolishes both PRC1 and PRC2 recruitment by Xist RNA[10]. Consistently we observed complete loss of Xist-dependent H2AK119ub and H3K27me3 after 1 day of Xist RNA induction in XistΔPID mESCs, as determined by calibrated allelic ChIP-seq (Supplementary Fig. 6b). Allelic ChrRNA-seq performed 1 day after Xist RNA induction demonstrated abrogation of Xist-mediated silencing across the chromosome (Fig. 6a, b). Interestingly, silencing was abrogated to a greater degree after 6 days of Xist RNA induction in differentiating conditions (Fig. 6c–e).

We went on to investigate if absence of Polycomb affects Xist-mediated silencing of all genes to an equivalent extent. Genes with pre-existing H3K27me3 compared with genes with a pre-existing active chromatin signature (classified by ChromHMM analysis) showed a greater reliance on Polycomb, both in Xist-dependent silencing in iXist-Chr3 and iXist-ChrX cells (Fig. 6f). To further investigate this observation we examined the acquisition of Polycomb-mediated histone modifications, H2AK119ub and H3K27me3 at the different groups based on chromatin signature. We observed that gain of Polycomb histone modifications was significantly higher at genes and regions classified as having pre-existing H3K27me3 chromatin in mESCs (Fig. 6g–j, Supplementary Fig. 7a–f). This, however, was not evident in the ChrX XistΔPID line. We conclude that enhanced accumulation of Polycomb modifications at pre-existing sites may contribute to preferential silencing of nearby genes, but that other variables also play a role in the observed differential effects.

**Integrated analysis of Xist-mediated silencing**. A comparison of the contribution made by different silencing pathways in iXist-Chr3 and iXist-ChrX lines after 1 day of Xist RNA induction is shown in Supplementary Fig. 8a, b. Based on this analysis we conclude that in both the X chromosome and autosomal models, Spen is the principal silencing pathway, with Polycomb (predominantly PRC1) also playing an important role. Rbm15/m6A-MTase complex and Lbr play a relatively minor role, detected only in the X chromosome model. These latter conclusions are supported by analysis of Xist RNA elements linked to binding/activity of the factors.

Our conclusions are further illustrated by principal component analysis (PCA) of individual biological replicates for iXist-Chr3 and iXist-ChrX after 1 day of Xist RNA induction (Fig. 7a, b). Thus, all uninduced controls, which are tightly clustered together, are clearly separated from induced datasets, in accord with the occurrence of Xist-mediated silencing. Notably, Xist-induced datasets for Spen knockout and the A-repeat deletion are close to or overlapping uninduced controls, indicating a dramatic silencing deficiency. Knockouts of genes encoding subunits of the Rbm15/m6A-MTase complex, LBR and associated Xist deletions (after Xist RNA induction), on the other hand, cluster closer to WT induced, with somewhat greater separation evident for iXist-ChrX compared to iXist-Chr3 cells. Pcgf3/5 and ΔPID datasets for iXist-Chr3 and iXist-ChrX, respectively, are further separated with a distinct direction relative to all the other induced WT and mutant datasets. This observation implies a unique role for the Polycomb system in the Xist-mediated silencing cascade.

To further depict relative silencing deficiencies we performed hierarchical clustering of allelic silencing (induced–uninduced), for datasets in iXist-ChrX cells (Fig. 7c). The results emphasise the dramatic and equivalent abrogation of silencing in Spen null and A-repeat deletion lines. Of the other pathways, Polycomb, predominantly PRC1-linked H2AK119ub (inferred from deletion of Xist PID region), is the most important factor, with Rbm15/

m6A-MTase and LBR playing a relatively minor role (Fig. 7c, d). LBR/LBS and Rbm15/m6A-MTase broadly impact silencing of all genes to an equivalent degree, but Polycomb has a more pronounced effect towards specific subsets of genes, notably those that are normally silenced most efficiently and associated with pre-existing H3K27me3 chromatin (Fig. 7c, d). This tendency was in fact also observed for the *Lbr* knockout cell lines, but to a much weaker degree (Fig. 7d). In most cases the different silencing pathways do not differentially affect silencing in relation to 2D/3D proximity to *Xist* (Fig. 7e, Supplementary Fig. 8c). However, A-repeat and/or Spen-independent silencing showed a weak preference with proximity to *Xist* (Fig. 7e, Supplementary Fig. 8c). This may indicate a defect in Xist RNA spreading/propagation, although it needs to be considered that there are only a small number of genes in the group showing partial silencing.

## Discussion

In this study we provide insights into the interplay and relative importance of different pathways required for the establishment of chromosome silencing by Xist RNA. Through the use of two mESC model systems we were able to gauge the contribution of the key silencing pathways in the context of Xist RNA action on the X chromosome and on an autosome. By using a sensitive approach to assay silencing chromosome-wide, we were able to determine how individual factors act on different subsets of genes.

Our results show that recruitment of the RBP Spen by the Xist A-repeat is the principal pathway for establishment of Xist-mediated silencing, affecting most genes to an equivalent degree. It was recently reported that deletion of HDAC3, which functions downstream of Spen in histone deacetylation, similarly has widespread effects on gene silencing[16]. We nevertheless found that some genes undergo a degree of silencing in the absence of Spen/A-repeat. Consistent with this observation, a recent study, which utilised allelic RNA-seq, also reported residual Xist-mediated silencing in response to constitutively driven expression of A-repeat-deleted Xist RNA in mouse extraembryonic tissues[39]. The degree of silencing observed in this study was higher than we find in the ES cell models, possibly reflecting differences in the level of Xist RNA and/or developmental stage/cell type. In common with our findings, weakly expressed genes were more resistant to loss of the A-repeat.

The fact that there is only very limited silencing following Spen/A-repeat loss of function could suggest that other pathways, Polycomb, and to a lesser degree, Rbm15/m6A-MTase and LBR, function downstream of Spen. However, a potentially confounding factor is that Spen/A-repeat deletion reduces Xist RNA levels. The basis for this effect is not clear but may relate to previous findings, indicating that the spread of Xist RNA from early nucleation sites is abrogated when the A-repeat is deleted[33]. A further caveat is that we only assayed silencing in Spen/A-repeat null mESCs up to 6 days after Xist RNA induction in differentiating mESCs, and it is possible that gene silencing increases at later timepoints. In support of this possibility we found that the rate of silencing in WT iXist-ChrX cells varies considerably for different genes, and even after 6 days of Xist RNA induction, gene silencing is in many cases incomplete.

A surprising conclusion from our study is that the Rbm15/m6A-MTase complex, previously proposed to be a key factor for Xist-mediated silencing[13,19], plays only a minor role. Thus, we observed a small reduction in silencing efficiency in the iXist-ChrX model and no effect in iXist-Chr3 cells. Interpretation of our observations is somewhat confounded by the fact that deletion of the m6A-MTase complex subunits does not entirely abrogate m6A levels. We speculate that this is attributable to various factors,

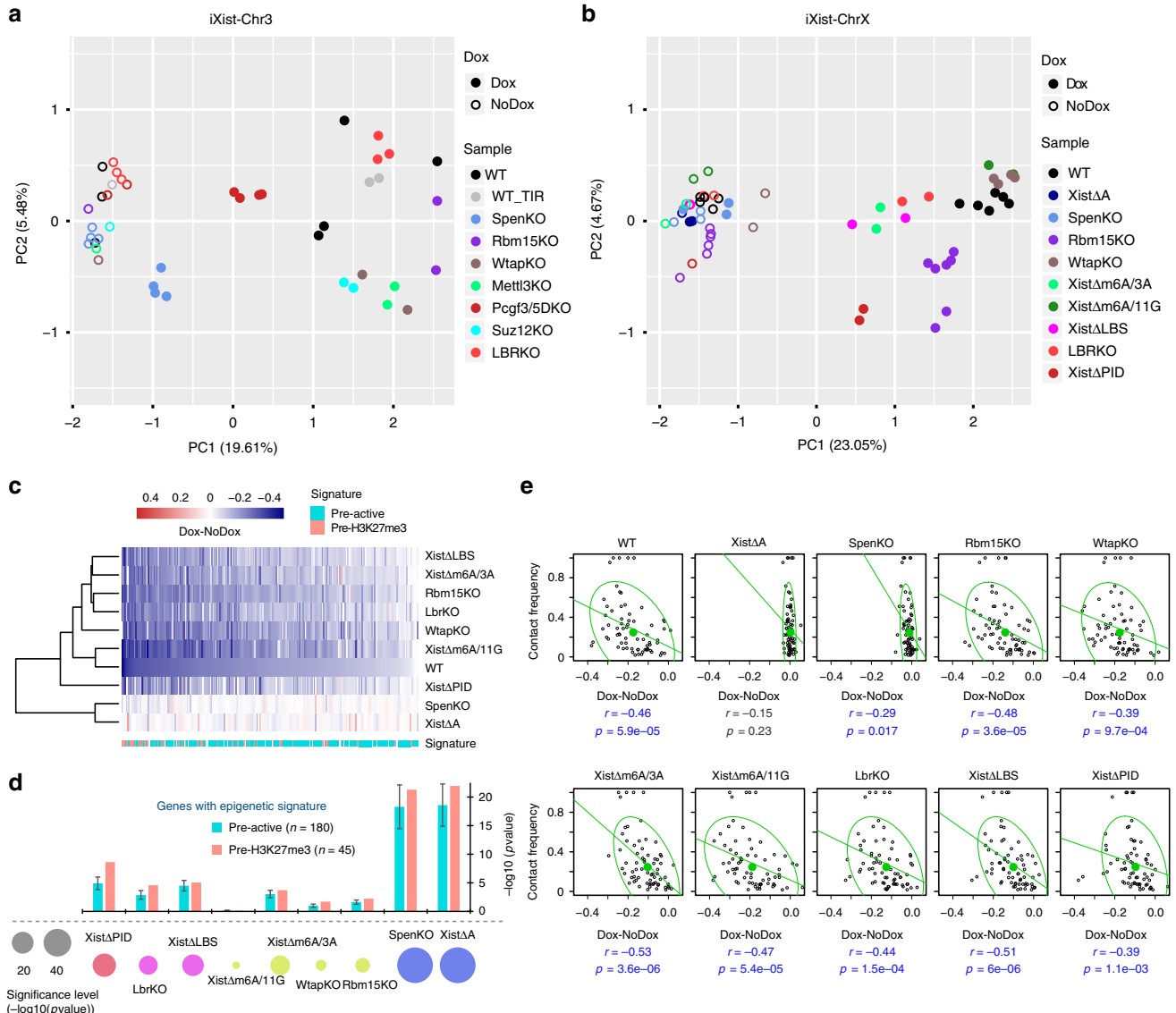

**Fig. 7** Integrated analysis of key pathways in Xist-mediated silencing. **a**, **b** PCA analysis of biological replicates of ChrRNA-seq analysis for all uninduced and Xist-induced experiments in iXist-Chr3 (**a**) and iXist-ChrX (**b**) for WT and mutants. **c** Cluster analysis and heatmap for silencing efficiency of ChrX genes in selected mutant mESCs, as indicated after 1 day of Xist RNA induction. Gene order is ranked by silencing efficiency in WT iXist-ChrX cells. Chromatin signature defined by ChromHMM is shown below. **d** Silencing levels in different mutants in iXist-ChrX cells illustrated as one-sided Wilcoxon rank-sum significance. Bar charts illustrate separate significance scores for genes classified according to pre-existing active (pre-active) and H3K27me3 (pre-H3K27me3) chromatin signatures, corrected for sample size. Error bars indicate significance range for 180 permutations. **e** Correlation analysis for topological distance and silencing efficiency for WT and selected mutant mESCs in iXist-ChrX model. Ninety-five per cent of the population is encircled. Spearman correlation and p values are indicated below

including redundancy (for example, Rbm15 and Rbm15b and possibly Mettl3 and Mettl14), and that specific subunits are not essential for m6A catalysis (for example the regulatory subunit Wtap). It is notable that complete loss of function of the m6A system in mESCs has never been reported, indicating that this would result in a strong cell lethality effect. While redundancy of Rbm15/m6A-MTase complex subunits and essentiality of the m6A system must be taken into account, our conclusion that this pathway plays only a minor role in Xist-mediated silencing is supported by analysis of deletions around the major m6A target region in Xist RNA. Possible reasons for the discrepancy in our findings compared to prior work include the use of different silencing assays (selected loci vs chromosome-wide), and acute (RNAi-mediated knockdown) vs chronic (mESC lines with CRISPR-mediated deletions) loss of function.

Similar to the Rbm15/m6A-MTase complex, deletion of *Lbr* had only a minor effect on Xist-mediated silencing in iXist-ChrX cells and no effect in iXist-Chr3 cells, again contrasting with previously reported findings[23]. We note that our observations are consistent with *Lbr* knockout mouse models which show no apparent female-specific phenotypes[24,25]. Deletion of the proposed LBR binding region, LBS, in Xist RNA led to a more prominent silencing deficiency. We conclude from this that LBS is important for other functions in addition to LBR binding. Discrepancies with prior studies[15,23] may again be linked to the use of different silencing and loss of function assays, as well as allelic vs non-allelic analysis.

The reason for the difference in silencing effects seen for Rbm15/m6A-MTase and LBR loss of function in iXist-ChrX compared to iXist-Chr3 systems is not known. One possibility is

that higher levels of Xist RNA in iXist-Chr3 cells, due to the presence of a multicopy Xist transgene, mask the relatively small effect of Rbm15/m6A-MTase and LBR deficiency. Alternatively, the Rbm15/m6A-MTase and LBR pathways may only be relevant in an X chromosome context, for example contributing to the higher silencing efficiency that has been observed to occur on the X chromosome relative to autosomes[9].

Our findings substantiate the principal role of the PID region and PRC1 in hierarchical recruitment of Polycomb complexes by Xist RNA, and the proposal that Xist-dependent recruitment of PRC2 is attributable solely to prior H2AK119ub deposition by PRC1 (refs. [10,29]). Our analysis further demonstrates that PRC1-mediated H2AK119ub is the principal determinant of Polycomb silencing mediated by Xist RNA. H3K27me3, catalysed by PRC2 contributed to a much lesser extent, although this effect did increase at a later timepoint. The basis for the progressive silencing deficiency is not clear but may reflect changes in levels of H3K27me3 reader proteins in ESC differentiation reported in previous studies[40,41]. A further consideration is our recent finding that the PID region is required to recruit SmcHD1, a factor implicated in late stages of X inactivation[42]. This pathway, which is mediated by PRC1-dependent H2AK119ub, may account for why we observed a greater silencing deficiency in XistΔPID cells after 6 days compared with 1 day of Xist expression.

In this work we have analysed pathways that function in establishment of silencing at the onset of Xist expression. However, previous studies using XX mESCs have shown that there are some factors, for example, histone macroH2A[43], and SmcHD1 (ref. [44]), that are recruited to Xi in an Xist-dependent manner, but only after prolonged Xist expression/cell differentiation. Notably, SmcHD1 is important for maintaining silencing of a significant proportion of Xi genes[45–47]. Thus, in future work it will be important to define the relationship between early- and late-acting pathways in establishment vs maintenance of Xist-mediated silencing.

## Methods

**Experimental model**. ES cells were grown in Dulbecco's modified Eagle's medium (DMEM; Life Technologies) supplemented with 10% foetal calf serum (Seralab), 2 mM L-glutamine, 0.1 mM non-essential amino acids, 50 μM β-mercaptoethanol, 100 U/ml penicillin/100 μg/ml streptomycin (all from Life Technologies) and 1000 U/ml LIF (made in-house). Mitomycin C-inactivated mouse fibroblasts were used as feeders. All ES cells were grown in feeder-dependent conditions on gelatinised plates at 37 °C in a humid atmosphere with 5% $CO_2$. Xist expression was driven by a TetOn promoter induced by addition of 1.0–1.5 μg/ml of doxycycline (Sigma, D9891) for 1–6 days depending on the experiment. ES cell differentiation was achieved by LIF withdrawal from the medium and low-density cell plating.

**Generation of mESC lines**. To generate iXist-ChrX ESC lines, F1 2–1 XX mESC line (129/Sv-Cast/Ei, a gift from J. Gribnau) was consecutively modified using CRISPR-mediated genome engineering to replace one of the endogenous Xist promoters with tetracycline-inducible promoter (tetOP), and to introduce reverse tetracycline-controlled transactivator (rtTA) into the Tigre locus[31]. The construct for Xist promoter modification (pBS_Xist_loxN_Pgkneo-tetOP) contained 2.6 kb (5′) and 2.8 kb (3′) homology regions with the insertion of tetOP 41 bp upstream of the Xist TSS. The construct included a floxed neomycin resistance gene driven by the mouse Pgk promoter for positive selection of the targeted clones. The construct for targeting the Tigre locus (pBS_TIGRE_CAG_rtTA) contained 1 kb homology regions, surrounding CAG promoter-driven rtTA. Both targeting events were facilitated by co-transfection of the targeting plasmids together with specific Xist (Xist_TRE_gRNA1 and gRNA2) or Tigre (TIGRE_gRNA) sgRNAs. Cells were transfected using Lipofectamine 2000 (Life Technologies) according to the manufacturer's instructions. One microgram of each sgRNA was lipofected together with 1 μg of the targeting vector. After 12 h transfected cells were passaged to 90 mm gelatinised Petri dishes with feeders, and puromycin (1.5–2.5 μg/ml) was applied 24 h later. Cells were grown under puromycin selection for 2 days and then without puromycin for a further 8–10 days, until colonies were picked. For targeting of the Xist promoter, neomycin selection (Geneticin® (G-418); Thermo-Fisher) was additionally applied 48 h after lipofection, and was maintained until colonies were picked. Selected clones were screened and validated by Sanger sequencing (SS) of amplified genomic fragments (gPCR), by Southern blot hybridisation (SB), by Xist RNA-FISH (RNA-FISH) and immunofluorescence for

histone modifications (IF) after Xist doxycycline induction and karyotyped using 21XMouse MFISH (Zeiss Metasystem), following the manufacturer's instructions. Two clones were selected for further analysis; clone B2 carries tetOP on the *M.m. domesticus* allele and clone C7 has integration of tetOP on the *M.m.castaneus* allele.

For the generation of CRISPR/Cas9-mediated KO ES cell lines, cells were transfected with 0.5–1 μg of each sgRNA, and puromycin selection was applied 24 h after transfection for 48 h. For CRISPR-assisted homologous recombination, 0.5–1 μg of sgRNA was lipofected together with 1 μg of targeting vector. Cells were grown for 10–12 days, and positive clones validated. Full details for all mESC lines used in this study, sgRNA sequences, plasmids, as well as validation methods can be found in Supplementary Table 1. Oligonucleotides used for cloning sgRNAs as well as primers used for PCR and sequencing are listed in Supplementary Table 2.

**RNA-FISH**. Cells were plated on Superfrost Plus gelatinised slides (VWR) and grown overnight. After Xist induction (1, 3, or 6 days) the slides were washed twice with PBS and cells were permeabilised for 5 min in CSK buffer (100 mM NaCl, 300 mM sucrose, 3 mM $MgCl_2$, 10 mM PIPES) with 0.5% Triton X-100 (Sigma) on ice. Slides were rinsed briefly in PBS and fixed in 4% formaldehyde/PBS for 10 min on ice, followed by two washes in 70% ethanol. Slides were either stored in 70% ethanol at 4 °C until use or dehydrated (80%, 95%, 100% ethanol, 3 min each, RT) and air dried immediately before hybridisation with Xist probe. Xist probe was generated from an 18 kb cloned cDNA spanning the whole Xist transcript using a nick translation kit (Abbott Molecular, 7J0001) as previously described[13]. Directly labelled probe (1.5 μl) was co-precipitated with 10 μg salmon sperm DNA, 1/10 volume 3 M sodium acetate (pH 5.2), and 3 volumes of 100% ethanol. After washing in 75% ethanol, the pellet was dried, resuspended in 6 μl deionised formamide (Sigma), and denatured at 75 °C for 7 min before quenching on ice. Probe was diluted in 6 μl 2× hybridisation buffer (5× SSC, 12.5% dextran sulfate, 2.5 mg/ml BSA (NEB)), added to the denatured slides and incubated overnight at 37 °C in a humid chamber. After incubation, slides were washed three times with a solution of 2× SSC/50% formamide followed by three washes with 2× SSC in a water bath at 42 °C. Slides were mounted with Vectashield with 4,6-diamidino-2-phenylindole (DAPI) (Vector labs) and sealed with nail varnish. Cells were analysed on an inverted fluorescence Axio Observer Z.1 microscope using a PlanApo ×63/1.4 NA oil-immersion objective. Images were acquired using AxioVision software.

**Immunofluorescence**. Cells were plated on gelatinised slides or coverslips (13 mm diameter coverslips from VWR), at least a day before the experiment. On the day of the experiment, cells on slides were washed with PBS and then fixed with 2% formaldehyde for 15 min followed by 5 min of permeabilisation in 0.4% Triton X-100. Cells were briefly washed with PBS before blocking with a 0.2% PBS-based solution of fish gelatine for three washes of 5 min each. Primary antibody dilutions were prepared in fish gelatine solution with 5% normal goat or normal donkey serum depending on the secondary antibodies used. A complete list of antibodies used in the study is shown in Supplementary Table 3. Cells on slides were incubated with primary antibodies for 2 h in a humid chamber at room temperature. Slides were washed three times in fish gelatine solution. Secondary antibodies were diluted 1:400 in fish gelatine solution and incubated with cells on slides for 30 min in a humid chamber at 37 °C. After incubation, slides were washed twice with fish gelatine and one time with PBS before mounting using Vectashield mounting medium with DAPI. Excess mounting medium was removed and the coverslips were sealed to slides using nail varnish. Cells were analysed on an inverted fluorescence Axio Observer Z.1 microscope using a PlanApo ×63 /1.4 NA oil-immersion objective. Images were acquired using AxioVision software.

**Auxin-inducible degradation**. For the degron-mediated degradation of PCGF3-AID-EGFP, indole-3-acetic acid (IAA, chemical analogue of auxin; Sigma, I5148) was added to the ES medium to a final concentration of 500 μM for 12 h prior to Xist induction for all experiments. IAA was maintained in the media at all times after initial addition. Degradation of PCGF3-AID-EGFP was monitored by western blot and IF.

**Nuclear extract**. Cell pellets were washed with PBS and resuspended in 10 packed cell volume (PCV) buffer A (10 mM HEPES-KOH pH 7.9, 1.5 mM $MgCl_2$, 10 mM KCl, with 0.5 mM DTT, 0.5 mM PMSF, and complete protease inhibitors (Roche) added fresh). After a 10 min incubation at 4 °C, cells were collected by centrifugation (1500g, 5 min, 4 °C) and resuspended in 3 PCV of buffer A + 0.1% NP40 (Sigma). After another 10 min incubation at 4 °C, nuclei were collected by centrifugation (400g, 5 min, 4 °C) and resuspended in 1 PCV buffer C (250 mM NaCl, 5 mM HEPES-KOH (pH 7.9), 26% glycerol, 1.5 mM $MgCl_2$, 0.2 mM EDTA-NaOH, pH 8.0 with complete protease inhibitors (Roche) + 0.5 mM DTT added fresh). Five molar NaCl was added drop-wise to bring the concentration to 350 mM and the mixture was incubated for 1 h at 4 °C with occasional agitation. After centrifugation (16,100g, 20 min, 4 °C), the concentration of the supernatant was quantified using the Bio-Rad Bradford assay and stored at −80 °C until use.

**Western blot**. Protein from nuclear extract was used for all experiments. Samples were separated on a polyacrylamide gel and transferred onto PVDF membrane by semi-dry transfer (15 V for 50 min). Membranes were blocked by incubating them

for 1 h at room temperature in 10 ml TBS, 0.1% Tween (TBST) with 5% w/v Marvel milk powder. Blots were incubated overnight at 4 °C with the primary antibody (see Supplementary Table 3), washed four times for 10 min with TBST and incubated for 40 min with secondary antibody conjugated to horseradish peroxidase. After washing four times for 5 min with TBST, bands were visualised using ECL (GE Healthcare). Original western blot images are available from the Mendeley Data depository from this link [https://doi.org/10.17632/52pjcxy486.1]

**Chromatin RNA-seq.** Chromatin RNA was extracted from one confluent 15 cm dish of mESCs. Briefly, cells were trypsinised and washed in PBS. Cells were lysed on ice in RLB (10 mM Tris pH 7.5, 10 mM KCl, 1.5 mM MgCl$_2$, and 0.1% NP40), and nuclei were purified by centrifugation through a sucrose cushion (24% sucrose in RLB). The nuclei pellet was resuspended in NUN1 (20 mM Tris pH 7.5, 75 mM NaCl, 0.5 mM EDTA, 50% glycerol), then lysed with NUN2 (20 mM Hepes pH 7.9, 300 mM, 7.5 mM MgCl$_2$, 0.2 mM EDTA, 1 M Urea). Samples were incubated for 15 min on ice then centrifuged at 2800$g$ to isolate the insoluble chromatin fraction. The chromatin pellet was resuspended in TRIzol by passing multiple times through a 23 gauge needle. Finally chromatin-associated RNA was purified through standard TRIzol/chloroform extraction followed by isopropanol precipitation. Samples were then treated with Turbo DNAse, and 500 ng–1 μg of RNA was used for library preparation using the Illumina TruSeq stranded total RNA kit (RS-122-2301). Libraries were quantified by qPCR with KAPA Library Quantification DNA standards (Kapa Biosystems, KK4903). Biological replicates for each experiment are listed in Supplementary Table 4. The libraries were pooled and 2 × 81 paired-end sequencing was performed using Illumina NextSeq500 (FC-404-2002).

**Native ChIP-seq.** For calibrated native ChIP (H3K27me3 and H2AK119ub), 40 million ES cells and 10 million Drosophila SG4 Cells were carefully counted using a LUNA-II Automated Cell Counter (Logos Biosystems), pooled and lysed in RSB (10 mM Tris pH 8, 10 mM NaCl, 3 mM MgCl$_2$, 0.1% NP40) for 5 min on ice with gentle inversion. Nuclei were resuspended in 1 ml of RSB + 0.25 M sucrose + 3 mM CaCl$_2$, treated with 200 U of MNase (Fermentas) for 5 min at 37 °C, quenched with 4 μl of 1 M EDTA, then centrifuged at 5000 r.p.m. for 5 min. The supernatant was transferred to a fresh tube as fraction S1. The chromatin pellet was incubated for 1 h in 300 μl of nucleosome release buffer (10 mM Tris pH 7.5, 10 mM NaCl, 0.2 mM EDTA), carefully passed five times through a 27G needle and then centrifuged at 5000 r.p.m. for 5 min. The supernatant S2 fraction was combined with S1 as soluble chromatin extract. For each ChIP reaction, 100 μl of chromatin was diluted in Native ChIP incubation buffer (10 mM Tris pH 7.5, 70 mM NaCl, 2 mM MgCl$_2$, 2 mM EDTA, 0.1% Triton) to 1 ml and incubated with antibody (see Supplementary Table 3) overnight at 4 °C. For H2AK119ub ChIP, all buffers were supplemented with 10 mM of the deubiquitinase inhibitor $N$-ethylmaleimide (Sigma, E3876-5G). Samples were incubated for 1 h with 40 μl protein A agarose beads pre-blocked in Native ChIP incubation buffer with 1 mg/ml BSA and 1 mg/ml yeast tRNA, then washed for a total of four times in Native ChIP wash buffer (20 mM Tris pH 7.5, 2 mM EDTA, 125 mM NaCl, 0.1% Triton) and once in TE. The DNA was eluted with 1% SDS and 100 mM NaHCO$_3$, and was purified using the ChIP DNA Clean and Concentrator kit (Zymo Research).

Approximately 20–50 ng of ChIPed DNA was used for library prep using the NEBNext Ultra II DNA Library Prep Kit with NEBNext Single indices (E7645), and then further quantified by qPCR with KAPA Library Quantification DNA standards and SensiMix SYBR (Bioline, UK). The libraries were pooled and 2 × 81 paired-end sequencing was performed using Illumina NextSeq500.

**m6A-seq.** m6A-seq was based on the method by Dominissini et al.[36]. Briefly, total RNA was extracted from pre-plated ES cells under reducing conditions using the RNeasy kit (Qiagen) and on-column DNase treatment as per the manufacturers' instructions. RNA was fragmented by incubation for 5 min at 94 °C in thin-walled PCR tubes with fragmentation buffer (100 mM Tris-HCl, 100 mM ZnCl$_2$). Fragmentation was quenched using stop buffer (200 mM EDTA, pH 8.0) and incubation on ice, before ensuring the correct size (50–100 bp) using 1.5% agarose gel electrophoresis. Three hundred and seventy-five micrograms of total RNA was incubated with 12.5 μg anti-m6A antibody (Synaptic Systems, 202 003), RNasin (Promega), 2 mM VRC, 50 mM Tris, 750 mM NaCl and 5% Igepal CA-630 in DNA/RNA low-bind tubes for 2 h before m6A containing RNA was isolated using 200 μl Protein A agarose beads (Repligen) per IP (pre-blocked with BSA). About 6.7 mM m6A (Sigma) was used to elute RNA from the beads. Input and eluate samples were EtOH precipitated, quantitated and pooled as libraries generated using TruSeq Stranded total RNA LT sample prep according to the manufacturer's instructions, but excluding the fragmentation step. 50 bp or 75 bp single-end reads were obtained using Illumina NextSeq500 sequencer (Illumina, FC-404–2005).

**m6A HPLC-MS/MS quantification.** mRNA was prepared from pre-plated ES cells using two sequential rounds of purification using the Dynabeads mRNA Direct kit (ThermoFisher, 61011) as per the manufacturer's protocol. RNA samples were hydrolysed with DNA Degradase Plus™ (Zymo Research) supplied with 40 nM deaminase inhibitor erythro-9-Amino-β-hexyl-α-methyl-9H-purine-9-ethanol hydrochloride (Sigma-Aldrich) at 37 °C for 4 h. After hydrolysis, the enzymes were removed with Amicon Ultra-0.5 ml 10K centrifugal filters (Merck Millipore).

The HPLC-MS/MS analysis was carried out with 1290 Infinity LC Systems (Agilent) coupled with a 6495B Triple Quadrupole Mass Spectrometer (Agilent). A ZORBAX Eclipse Plus C18 column (2.1 × 150 mm, 1.8-μm; Agilent) was used. The column temperature was maintained at 40 °C, and the solvent system was water containing 10 mM ammonium acetate (pH 6.0, solvent A) and methanol (solvent B) with 0.4 ml/min flow rate. The gradient was: 0–5 min; 0 solvent B; 5–8 min; 0–5.63% solvent B; 8–9 min; 5.63% solvent B; 9–16 min; 5.63–13.66% solvent B; 16–17 min; 13.66–100% solvent B; 17–21 min; 100% solvent B; 21–24.3 min; 100–0% solvent B; 24.3–25 min; 0% solvent B. The dynamic multiple reaction monitoring mode (dMRM) of the MS was used for quantification. The source-dependent parameters were as follows: gas temperature 230 °C, gas flow 14 L/min, nebuliser 40 psi, sheath gas temperature 400 °C, sheath gas flow 11 L/min, capillary voltage 1500 V in the positive ion mode, nozzle voltage 0 V, high pressure RF 110 V and low pressure RF 80 V, both in the positive ion mode. The fragmentor voltage was 380 V for all compounds.

Nucleosides were quantified using the precursor ion to product ion $m/z$ transitions of 268 → 136 for A, and 282 → 150 for m6A. The retention time for A and m6A is 14.9 ± 2 and 18.4 ± 2 min, respectively. The cell accelerator voltage was set at 4 V and the collision energy was set at 8 V for A and 16 V for m6A. Quantification of the m6A/A ratio was calculated using the calibration curves generated with nucleoside standards running within the same run.

**iXist-Chr3 integration site identification.** To determine the integration locus of the inducible Xist transgene in Chromosome 3, over 500 million paired-end reads from all ChIP-seq experiments were combined and mapped to the mm10 mouse genome. All read pairs which could not be mapped were then used for further analysis. Paired reads where one mapped to Chr3 and the other to the plasmid containing the Xist transgene were identified and quantified. The region in Chr3 that contained the highest concentration of these chimeric reads was Chr3:125.8097–125.8145MB. The integration site was also confirmed by using combined reads from previously generated ATAC-seq data[10].

**Chromatin state segregation by ChromHMM.** Using mESC ENCODE ChIP-seq data for various histone modifications and transcription factors, we defined 12 chromatin states (strong enhancer, promoter, gene body, etc) by ChromHMM (v1.11)[35] following the instructions given at [https://github.com/guifengwei/ChromHMM_mESC_mm10] (state 1: CTCF-binding; state 2: Intergenic_Region; state 3: Heterochromatin; state 4: Enhancer; state 5: RepressedChromatin; state 6: BivalentChromatin; state 7: ActivePromoter; state 8: StrongEnhancer; state 9: TranscriptionTransition; state 10: TranscriptionElongation/GeneBody; state 11: Weak/poised_Enhancer; state 12: LowSignal/RepetitiveElements). ChromHMM was used to determine the chromatin state of each gene based on 2 kb flanking either side of the TSS. When multiple transcripts were annotated, we chose the TSS used for the highest number of unique transcripts, or one at random if multiple TSSs were equally frequent. When multiple chromatin states were present in the TSS ± 2 kb region, the predominant state was assigned to the gene.

**RNA-seq data analysis.** The following analysis strategy was used for Chromatin RNA-seq and mRNA-seq. The RNA-seq data mapping pipeline was similar to our previous study[10]. Briefly, the raw fastq files of read pairs were first mapped to an rRNA build by bowtie2 (v2.3.2)[48] and rRNA-mapped reads discarded. The remaining unmapped reads were aligned to the "N-masked" genome (from mm10 coordinates) with STAR (v2.4.2a) using parameters "–outFilterMultimapNmax 1 –outFilterMismatchNmax 4 –alignEndsType EndToEnd" for all the sequencing libraries[49]. Unique alignments were retained for further analysis. We made use of 23,005,850 SNPs between Cast and 129S genomes and employed SNPsplit (v0.2.0; Babraham Institute, Cambridge, UK) to split the alignment into distinct alleles (Cast and 129S) using the parameter "–paired". The (allelic) read numbers were counted by the program featureCounts (-t transcript -g gene_id -s 2)[50] and the alignments were sorted by Samtools[51]. BigWig files were generated by Bedtools[52] and visualised by IGV[53] or UCSC Genome Browser. For biallelic analysis, counts were normalised to one million mapped read pairs (as CPM) by the edgeR R package. Genes with at least 10 SNP-covering reads across all the samples were further taken to calculate the allelic ratio of Xi/(Xi + Xa), where Xi and Xa indicate inactive and active allele, respectively.

In the iXist-Chr3 XY cell line the inducible Xist transgene expresses from the Cast allele, whereas in the iXist-ChrX XX line the endogenous Xist from the 129S1 allele is induced (see Fig. 1a). For some purposes we also used data from ChrRNA-seq performed in a reciprocal cell line iXist-ChrX XX C7H8 in which endogenous Xist from the Cast allele is induced. Silencing is primarily quantified in all cell lines i.e. wild-type Xist, Xist mutants, and Xist-interacting factor KOs by the difference in allelic ratios between uninduced and induced samples, such that

$$\text{Gene silencing (z)} = \left[\frac{Xi}{Xi + Xa}\right]_{Dox} - \left[\frac{Xi}{Xi + Xa}\right]_{NoDox}$$

For further comparisons, we set thresholds of $z > -0.05$, $-0.05 < z < -0.2$, $z < -0.2$ to represent genes demonstrating weak/no, low, and high silencing respectively. Additionally, original levels of gene expression for each cell line were calculated as

FPKM by Cuffnorm (v2.2.1)[54], and used to further categorise genes into three graded expression groups (E1 < E2 < E3) with the same number of genes per group.

Data generated in Suz12 KO lines and corresponding wild-type iXist-Chr3 after 3 days in differentiation media were processed with the same pipeline as above. The *p* values were calculated by Student's *T*-test based on the three biological replicates with 0.05 as the significance cutoff, and additional Repression Score (RS) analysis was performed as our previous study[10].

PCA analysis was performed by R function prcomp and plotted with the R package ggplot2.

**Native ChIP-seq calibration and data analysis**. For ChIP-seq experiments quantitatively calibrated with Drosophila SG4 cells, raw fastq read pairs were mapped to the "N-masked" mm10 genome concatenated with the dm6 genome sequence by STAR (v2.4.2a)[49] using the same parameters as for RNA-seq data with the additional parameter "–alignIntronMax 10". Allelic split of mm10-mapped reads was conducted by SNPsplit as for RNA-seq. We then normalised the mm10-mapped reads to spiked-in Drosophila library size, and scaled down the WT sample to 10 million reads and the other calibrated samples by the same factor, for processing by Bedtools[52] into bedGraph files for normalised signal. All the parameters and normalisation intermediates can be found in the Source Data. We calculated absolute gain of H3K27me3 and H2AK119ub upon Xist induction by the formula $(Xi - Xa)_{Dox} - (Xi - Xa)_{NoDox}$ for further analysis and visualisation as BigWig files. Custom scripts (ExtractInfoFrombedGraph_AtBed.py) were used to extract values from sorted bedGraph files for different regions (e.g. 250 kb windows or ChromHMM annotations), with signal from regions of different sizes comparable by FPKM. Metagene profiles for pre-active and pre-H3K27me3 gene sets were generated by DANPOS[55] with "–bin_size 50" and otherwise default parameters. To calculate the significance of Polycomb mark gain between pre-active and pre-H3K27me3 genes, we computed the average signal (by UCSC utility: bigWigAverageOverBed) from gene body regions, respectively. The signal of Polycomb gain across the intragenic regions of each group were compared, and significant levels were calculated by Wilcoxon rank-sum test.

**m6A-seq data analysis**. For conventional m6A-seq data, first we removed the rRNA reads computationally by mapping the single-end reads to the mouse rRNA build with Bowtie2 (ref. [48]). The remaining unmapped reads were then aligned to mm10 genome by STAR (v2.4.2a)[49] with "–single-end" mode. BigWig files were generated by Bedtools[52] and normalised to 10 million mapped reads (Generate_BigWig_from_RNA_seq_Bam_mm10.sh), and then visualised in IGV or UCSC.

**HiChIP data analysis and virtual 4C**. To integrate the data of higher order chromosome structure in mESCs, we downloaded two biological replicates of cohesin HiChIP-seq data in mESCs[34] and processed data according to the HiC-Pro pipeline[56]. The Xist loci in iXist-Chr3 and iXist-ChrX were used as viewpoints for virtual 4C visualisation at 1 MB resolution across Chr3 or ChrX. For comparison we also extracted calibrated ChIP-seq signal of H3K27me3 and H2AK119ub gain averaged at 1 MB resolution across Chr3 or ChrX. The correlations between Xist virtual 4C and ChIP-seq or RNA-seq were calculated using these 1 MB windows across the chromosome excluding the regions containing Xist locus as well as its flanking 10 MB.

**Machine learning**. The random forest classifier in the h2o R package (https://github.com/h2oai/h2o-3) was implemented to train models (key parameters: ntrees = 50, keep_cross_validation_predictions = T, nfolds = 5) for predicting the silencing efficiency ("High" vs "Low" silenced on the basis of above or below the median of silencing degree; i.e. 1 vs 0 in the training, validation and testing matrices) in both iXist-Chr3 and iXist-ChrX C7H8 line (Xist was expressed in Cast allele under the engineered TetOn promoter). Three types of features based on different characteristics of genes were used as inputs. (1) Gene expression. Original gene expressed level (FPKM), which was calculated from Cuffnorm program (v2.2.1)[54]. Each gene was assigned a value from 0 to 1 using two different strategies. In the first method, the highest expressing gene was normalised to one and the rest scaled down accordingly. In the second method, genes were ranked based on expression level, the highest expressing gene was assigned 1 and the lowest assigned 0 with the rest evenly distributed between 0 and 1. (2) Distance from the Xist locus for each gene. Genomic distance (2D) from the Xist locus was calculated as $1 - \left| \frac{Xist\_locus - gene\_locus}{Xist\_locus} \right|$. Additionally, topological distance to Xist locus (3D) was calculated from Xist virtual 4C[34], described as above, then normalised to the highest value to scale down to range 0–1. (3) Epigenetic state. Given that gene expression is tightly coordinated by epigenetic setting, we extracted and calculated the proportion of each chromatin state from 4 kb surrounding the TSS regions covering all the isoforms, characterised by ChromHMM[35] as above. For genes containing multiple TSSs, the proportion of each chromatin state within a composite of 4 kb TSS regions was used and weighted according to frequency of TSS usage. The epigenetic state encompassed 12 features, each of ranging from 0 to 1, representing the fraction of each chromatin state in the TSS regulatory region.

**Correlation between gene silencing and virtual 4C data**. The topological distance to Xist locus was calculated from the cohesin HiChIP-seq data[34], and then scaled down to range 0–1 as described above. The degree of silencing was calculated for each 1 MB genomic window as the ratio of expressed reads from this window $(Xi/(Xi + Xa))$. The raw counts for each 1 MB window were calculated by featureCounts[50] without strand specificity. For each factor knockout or Xist region deletion, biological replicates were averaged (Supplementary Table 4), and silencing scores $\left( \left[ \frac{Xi}{Xi+Xa} \right]_{Dox} - \left[ \frac{Xi}{Xi+Xa} \right]_{NoDox} \right)$ across the chromosome were calculated as above. Only 1 MB genomic windows that contain both valid virtual 4C value and at least 20 allelic reads were considered. The correlation was calculated in R by the "Spearman" method.

**Silencing preference on pre-active and pre-H3K27me3 marked genes**. We examined the silencing deficiency in gene sets with different pre-existing chromatin epigenetic signatures, i.e. pre-active and pre-H3K27me3 marks. Based on the aforementioned ChromHMM annotation, 180 and 45 genes are annotated as pre-active and pre-H3K27me3 marked genes, respectively. We employed one-sided Wilcoxon rank-sum tests to examine significance with the hypothesis that factor knockout or Xist sequence deletion will cause silencing deficiency. To eliminate the discrepancy in gene number between the two sets, we permutated 180 times by random sampling 45 genes from pre-active gene sets, and then performed the statistics. The range of significance level was the standard deviation of these 180 statistical analyses (Fig. 7).

**Reporting Summary**. Further information on research design is available in the Nature Research Reporting Summary linked to this article.

## Data availability

High-throughput sequencing data (ChrRNA-seq, m6A-seq, ChIP-seq) generated in this study have been deposited at GEO under accession number GSE119607. Original unprocessed gel images in this manuscript have been deposited in Mendeley Data and are available following this link [https://doi.org/10.17632/52pjcxy486.1].

Raw data used for ChIP-seq calibration and raw data underlying all reported averages in graphs and charts are provided as a Source Data File. All other relevant data supporting the key findings of this study are available within the article and supplementary files or from the corresponding author upon reasonable request.

## Code availability

Scripts used for analysis are available at [https://github.com/guifengwei].

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

## Acknowledgements

We thank Amanda Williams for assistance with NGS sequencing and Andrew Bassett for CRISPR/Cas9 sgRNA design for Wtap and Rbm15 mutagenesis; Neil Blackledge for providing the Tir targeting construct; Martin Houlard for providing TIGRE gRNA and HR construct; Michal Gdula for preliminary analysis of LBR datasets. This work was funded by grants to N.B. from the Wellcome Trust (103768) and the European Research Council (340081). C.-X.S. is supported by the Ludwig Institute for Cancer Research, Cancer Research UK (C63763/A26394), NIHR Oxford Biomedical Research Centre, and Conrad N. Hilton Foundation. Y.B. is supported by China Scholarship Council.

## Author contributions

T.B.N., G.W., H.C. and G.P. contributed to conceiving the study, experimental work, data analysis and manuscript preparation. J.S.B., T.Z., M.A., B.B., B.M., E.J.C., I.A.R., Q.P. and Y.B. contributed to experimental work. C.S. contributed to analysis of m6A mass spectrometry data. N.B. contributed to conceiving the study and writing the manuscript.

## Additional information

**Competing interests:** The authors declare no competing interests.

