## [Peer Review File · Nature Communications]

Reviewers' comments:

Reviewer #1 (Remarks to the Author):

In this manuscript by Nesterova and colleagues, and entitled "Systematic allelic analysis defines the interplay of key pathways in X-chromosome inactivation", the authors are dissecting the different pathways already described to be involved in XCI. To do so, they are using a collection of mutants in two different polymorphic ESC systems: one female line with an endogenous inducible Xist (iXistChrX) and one male line with an inducible Xist transgene on chromosome 3. These two systems allow the authors to look at early gene silencing upon Xist induction in either the X-chromosome or an autosome context.

This manuscript represents a large amount of work and tens of CRISPR mutants done in both models (iXistChr3 and iXistChrX) and highlights the preponderant role of Spen and Polycomb in XCI. However, the study lacks novel insights, do not really dissect the interplay between the different pathways as well as their function in X-linked gene silencing initiation and/or maintenance, all of that greatly dampened my enthusiasm.

Major comments:

- 1) This paper is lacking indispensable controls such as validation of the KO cell lines by western blot and/or qPCR, quantification of nearly all the RNA-FISH experiments and statistical analysis of several datasets. For example, in the Wtap KO lines, could the absence (or very subtle) phenotype on XIC be explained by a persistence of the protein?
- 2) It is unclear how the authors have determined the "pre-H3k27me3 associated genes". Are these genes only sharing H3K27me3 enrichment before Xist induction?
- 3) Authors claimed that the most efficiently silenced genes (both on chr3 and X) are the lowly expressed one, Figure 1 i. Is that statistically significant? Which test has been used?
- 4) The Spen mutant is showing the most drastic phenotype both on chr3 and chrX, very similar to Xist Δ A. What are the genes that are still silenced in absence of Spen? Are they closer to Xist, which features do they share? Could they be silenced independently of H3K27me3 as hypothesised for the early silenced genes in imprinted XCI (Inoue et al, 2018). And more importantly, the authors detect a very important decrease of Xist RNA in Spen KO (Sup.Fig2b, c, missing p-values). How could that be explain? Is the phenotype of Spen KO linked to the absence of Spen or to the reduce level of Xist RNA?
- 5) All the analysis on Wtap, Rbm15 and Mettl3 function are misleading as we do not know if it is proper KO ES cells as well as if the pathway is truly affected. For example, the m6Aseq data show persistence of a peak at the Xist locus in both iChr3 and iChrX cell lines. From this observation, how could any conclusion be drawn on the role of m6A MT pathway? Moreover, the authors claimed that mutations in Rbm15, Mettl3 and Wtap do not affect Xist RNA level (Sup. Fig 2b). Do the authors have any statistical analysis to support their claim as the Xist level seems affected at least in Chr3 context.
- 6) Analysis of the PRC1 and PRC2 complexes is important to better understand their interplay in XCI. In the Suz12 mutant, genes affected by PRC2 loss are more highly expressed before XCI initiation (stat?). These genes are more expressed in mESCs in absence of Suz12. Are these genes the most expressed also in a WT situation?

- 7) In Figure 6, are H3K27me3 and H2AK119ub tracks really significantly different between pre-active and pre-H3K27me3 genes? There is 2 times less genes for the pre-H3K27me3 group. Could that explain the noisier tracks?
- 8) Moreover, the authors analysed Polycomb involvement in ChrX context using Xist Δ PID mutant and showed that silencing defect is higher after 6 days of Xist induction (Figure 6E). Is that due to a maintenance defect rather than initiation defect? Same question for the role of the LBR and m6A MT complex?

Minor comments:

- 1) In Sup Figure 1a, the authors test Xist induction by adding Dox in iXist-ChrX. RNA-FISH should also be done in - Dox to test the leakiness of the system.
- 2) The authors should provide a table with the allelic scores of X-linked genes in the different mutant contexts. It would be useful to compare X-linked gene behaviours in the different mutant context.
- 3) Figure1 g,h: what are the different states?
- 4) Page 15: In Xist Δ PID the silencing is affected but not abrogated.
- 5) Missing p-values Fig 6g-j, page 16.

Reviewer #2 (Remarks to the Author):

In their article entitled "Systematic allelic analysis defines the interplay of key pathways in X chromosome inactivation", Brockdorff and colleagues systematically analyzed Xist-mediated silencing in ES cell-based models. They show that the synergistic activity of the Spen and Polycomb pathways is chiefly responsible for establishment of repression, and that – contrary to previous studies – LBR and the RBM15/m6A-methyltransferase complex make at most a minor contribution. In my review I was asked to focus on the contribution of m6A to repression. I carefully reviewed the experimental approaches, the results and their interpretation in the main text and discussion. I find the study to be very well designed, meticulously executed, including all appropriate controls, and carefully and appropriately interpreted. They tested the contribution of components of m6A methyltransferase complex (Rbm15, Mettl3, Wtap) and then reciprocally that of specific m6A sites on Xist to X-inactivation. Both lines of investigation support the conclusion that m6A plays a more minor role than previously thought, at least in this experimental system. There are several differences between this study and the previous ones which could potentially account for the differences in repression activity, and they are explicitly laid out in the discussion. However, I would emphasize more strongly in the discussion the possibility that the knockouts were incomplete as indicated by residual m6A levels. This means that the results should be taken with a grain of salt. Had the authors analyzed the contribution of the m6A binding protein YTHDC1 the picture would have been more complete. I support publication of the study in the current form given that the authors elaborate on the caveats in the discussion as I pointed out above.

Reviewer #3 (Remarks to the Author):

This manuscript by Nesterova et al. describes and compares the role of several factors in the X chromosome inactivation process. Previous work of the Brockdorff and other laboratories have implicated several chromatin remodelling factors to play a crucial role in this process, and in this study the individual role of Spen, Rbm15, WTAP, PRC1, PRC2 and many other factors have been investigated. For their studies they employ two inducible systems with a dox inducible Xist gene located on chromosome 3 in male cells, or an inducible endogenous Xist locus in female ES cells. Their studies indicate Spen is the central player in the XCI process acting through the A repeat in the 5' region of Xist. Other factors such as WTAP and LBR appear to play less important or redundant roles in the XCI process. Overall this is a very good and thorough study of importance for the XCI field and beyond. I have a few suggestions and questions to improve the manuscript. Regarding the induction the authors should explain more thoroughly how the experiment is performed. Is dox induction started prior to differentiation and for how long? Also in comparison to previous studies of several laboratories using ES cells as model system silencing in this present study appears to happen very quickly.

Part of the X chromosome appears to be non-informative. Why is this were these F1 or N1 129:cas ESCs?

In Suppl. Figure 2f the authors claim a correlation between silenced genes in Spen KO and delta-A cells. To me this correlation is not very clear at all despite a significance value for the hypergeometric test, why not use pearson or spearman here?

Why is the data represented Xi vs Xi+Xa in Figure 1, whereas in the other Figures dox- vs dox + is shown. This needs explanation.

The authors claim that 'silencing was abrogated to a greater degree after 6 days of Xist RNA induction in differentiating conditions (Fig. 6c-e)', however the relative ratio vs control remains the same, so what does this mean?

It is unfortunate that the Suz12 KO was only examined in the chr 3 context, but I guess this is a consequence of how in real time the order of experiments was performed.

We thank the reviewers for their thoughtful comments and suggestions.

Reviewer #1

In this manuscript by Nesterova and colleagues, and entitled “Systematic allelic analysis defines the interplay of key pathways in X-chromosome inactivation”, the authors are dissecting the different pathways already described to be involved in XCI. To do so, they are using a collection of mutants in two different polymorphic ESC systems: one female line with an endogenous inducible Xist (iXistChrX) and one male line with an inducible Xist transgene on chromosome 3. These two systems allow the authors to look at early gene silencing upon Xist induction in either the X-chromosome or an autosome context. This manuscript represents a large amount of work and tens of CRISPR mutants done in both models (iXistChr3 and iXistChrX) and highlights the preponderant role of Spen and Polycomb in XCI. However, the study lacks novel insights, do not really dissect the interplay between the different pathways as well as their function in X-linked gene silencing initiation and/or maintenance, all of that greatly dampened my enthusiasm.

Major comments:

1) This paper is lacking indispensable controls such as validation of the KO cell lines by western blot and/or qPCR, quantification of nearly all the RNA-FISH experiments and statistical analysis of several datasets. For example, in the Wtap KO lines, could the absence (or very subtle) phenotype on XIC be explained by a persistence of the protein?

We have added the validation data (western blot, Southern blot, RNA-seq, RNA-FISH) for all of the KO lines described in the revised manuscript.

2) It is unclear how the authors have determined the “pre-H3K27me3 associated genes”. Are these genes only sharing H3K27me3 enrichment before Xist induction?

We determined the “pre-H3K27me3 associated genes” using ChromHMM segmentation (see our Methods for interpretation of chromatin states). This is derived using published ES cell data and therefore corresponds to the state prior to Xist induction. Briefly, for these genes the chromatin state in the 4kb region surrounding gene promoter is state 5 and state 6 (predominant H3K27me3 signal). These genes are lowly expressed before Xist induction.

3) Authors claimed that the most efficiently silenced genes (both on chr3 and X) are the lowly expressed one, Figure 1 i. Is that statistically significant? Which test has been used?

The statistical analysis (**one-way ANOVA test**) has been performed and the significance level has been added to the panel. We thank the reviewer for the suggestion.

4) The Spen mutant is showing the most drastic phenotype both on chr3 and chrX, very similar to XistDA. What are the genes that are still silenced in absence of Spen?

The only feature that we could define for genes that are silenced better in Spen KO is that they are low expressed genes and/or genes associated with repressive chromatin marks (H3K27me3).

Are they closer to Xist, which features do they share?

There is a weak preference for proximity to the Xist locus. This is mentioned at the end of the results section.

Could they be silenced independently of H3K27me3 as hypothesised for the early silenced genes in imprinted XCI (Inoue et al, 2018).

Taking the early silenced gene *Chic1* as a representative example (Inoue et al, 2018), it is silenced neither in *Xist*ΔA nor *Spen*KO (see **Reviewer Figure 1**).

Reviewer Figure 1

And more importantly, the authors detect a very important decrease of *Xist* RNA in *Spen* KO (Sup.Fig2b, c, missing p-values). How could that be explain?

Xist RNA levels were in some instances from only two experiments, hence no p-values. However, in the case of *Spen* knockout, we have seen reduced levels of *Xist* RNA in several independent cell lines/experiments (>5) and are therefore confident that this is a true effect. In fact reduced *Xist* RNA levels in *Spen* knockouts was also reported in a published study (Montfort et al, 2015). We commented on this observation in the discussion section of our paper, suggesting a possible link to the reported effect of deleting the A-repeat on properties of *Xist* RNA (localisation/stability?) (p18, para 3).

Is the phenotype of *Spen* KO linked to the absence of *Spen* or to the reduce level of *Xist* RNA?

We cannot at present define the relative contribution of loss of silencing activity and reduced *Xist* RNA levels although a recently published study on the effects of deleting HDAC3, which functions downstream of *Spen* in gene silencing, reports similar widespread effects on X-linked genes, indicating that loss of silencing activity is a key determinant of the *Spen* knockout phenotype. We have added a reference to this recent paper in the discussion of our paper.

5) All the analysis on Wtap, Rbm15 and Mettl3 function are misleading as we do not know if it is proper KO ES cells as well as if the pathway is truly affected. For example, the m6Aseq data show persistence of a peak at the Xist locus in both iChr3 and iChrX cell lines. From this observation, how could any conclusion be drawn on the role of m6A MT pathway? Moreover, the authors claimed that mutations in Rbm15, Mettl3 and Wtap do not affect Xist RNA level (Sup. Fig 2b). Do the authors have any statistical analysis to support their claim as the Xist level seems affected at least in Chr3 context.

We provided m6A-MS data demonstrating significant reduction of m6A in Wtap and Mettl3 knockouts (supplementary Fig. 3d). Rbm15 knockout shows far less of an effect, probably because of redundancy with Rbm15b (Patil et al, Nature, 2016) and the fact that m6A complex can function independent of Rbm15 proteins. The fact that we do not see complete loss of m6A is consistent with prior studies using knockdown, and also with a similar CRISPR/Cas9 mediated mutation of Mettl3 (Batista et al, Cell Stem Cell, 2014). Whilst the reason for this is currently unknown (alternative redundant pathway?), our failure to derive ES cells in which the m6A system is completely lost is consistent with reports from others (Patil et al, 2016), and we believe reflects an essential role for m6A in cell viability. It was with this in mind that we performed experiments to delete the m6A target site in Xist RNA, which in our view more definitively shows the relatively minor contribution of this pathway in Xist-mediated silencing. We have added some further discussion concerning our analysis of the m6A system in response to comment from reviewer 2 (discussion, p19).

6) Analysis of the PRC1 and PRC2 complexes is important to better understand their interplay in XCI. In the Suz12 mutant, genes affected by PRC2 loss are more highly expressed before XCI initiation (stat?). These genes are more expressed in mESCs in absence of Suz12. Are these genes the most expressed also in a WT situation?

What we observed here is that the Suz12 sensitive genes have relatively higher original expression level before XCI initiation than those that are resistant. We called gene expression level in WT and Suz12 knockout cells by edgeR, and further calculated the $\log_2(\text{fold change})$ for each individual gene located on Chr3. The expression level of genes affected by PRC2 loss doesn't significantly alter upon Suz12 knockout (Wilcoxon rank-sum test), also true for genes that are resistant to PRC2 loss (**Reviewer Figure 2**). This result further indicates sensitivity to Suz12 loss is linked to Xist-induced silencing. We thank the reviewer for their comments and suggestions.

Reviewer Figure 2

7) In Figure 6, are H3K27me3 and H2AK119ub tracks really significantly different between pre-active and pre-H3K27me3 genes? There is 2 times less genes for the pre-H3K27me3 group. Could that explain the noisier tracks?

We thank the reviewer for pointing this out. To calculate the significance of Polycomb mark gain between pre-active and pre-H3K27me3 genes, we computed the average signal (by UCSC utility: bigWigAverageOverBed) from gene body regions, respectively. The signal of polycomb gain across the intragenic regions of each group were compared, and significance levels were calculated (Suppl. Fig. 7)..

For the metagene profile, the average values are taken for the plot. The noise results from use of a smaller number of genes/informative SNPs in pre-H3K27me3 genes.

8) Moreover, the authors analysed Polycomb involvement in ChrX context using XistDPID mutant and showed that silencing defect is higher after 6 days of Xist induction (Figure 6E). Is that due to a maintenance defect rather than initiation defect? Same question for the role of the LBR and m6A MT complex?

Our experiments do not allow us to definitively discriminate these possibilities – for this it would be necessary to investigate all possible intermediate and later timepoints, which is beyond the scope of the current study. However, in a collaborative study with the lab of Marnie Blewitt we found Xist PID region is required to recruit SmcHD1, a factor involved in late-stage silencing in XCI (this occurs via PRC1 mediated H2AK119ub1). This provides a possible rationale for the increased deficit in silencing seen at later timepoints (Jansz et al, Cell Reports, 2018). We didn't analyse later timepoints for mutations of m6A factors as these block ES cell differentiation, as reported previously. We have however analysed LBR and LBS mutants at 6 days, and observed if anything even less effect on silencing compared to early timepoints. We haven't included this data in the manuscript as it doesn't change our overall conclusions.

Minor comments:

1) In Sup Figure 1a, the authors test Xist induction by adding Dox in iXist-ChrX. RNA-FISH should also be done in - Dox to test the leakiness of the system.

We have provided this data as requested (supplementary Figure 1b).

2) The authors should provide a table with the allelic scores of X-linked genes in the different mutant contexts. It would be useful to compare X-linked gene behaviours in the different mutant context.

The data is in GEO database.

3) Figure1 g,h: what are the different states?

We have described the chromatin states in Methods.

4) Page 15: In XistDPID the silencing is affected but not abrogated.

Amended.

5) Missing p-values Fig 6g-j, page 16.

Amended. P-values were calculated by one-side Wilcoxon test, and added in Suppl. Fig. 7.

Reviewer #2

(Remarks to the Author):

In their article entitled "Systematic allelic analysis defines the interplay of key pathways in X chromosome inactivation", Brockdorff and colleagues systematically analyzed Xist-mediated silencing in ES cell-based models. They show that the synergistic activity of the Spen and Polycomb pathways is chiefly responsible for establishment of repression, and that – contrary to previous studies – LBR and the RBM15/m6A-methyltransferase complex make at most a minor contribution. In my review I was asked to focus on the contribution of m6A to repression. I carefully reviewed the experimental approaches, the results and their interpretation in the main text and discussion. I find the study to be very well designed, meticulously

executed, including all appropriate controls, and carefully and appropriately interpreted. They tested the contribution of components of m6A methyltransferase complex (Rbm15, Mettl3, Wtap) and then reciprocally that of specific m6A sites on Xist to X-inactivation. Both lines of investigation support the conclusion that m6A plays a more minor role than previously thought, at least in this experimental system. There are several differences between this study and the previous ones which could potentially account for the differences in repression activity, and they are explicitly laid out in the discussion. However, I would emphasize more strongly in the discussion the possibility that the knockouts were incomplete as indicated by residual m6A levels. This means that the results should be taken with a grain of salt. Had the authors analyzed the contribution of the m6A binding protein YTHDC1 the picture would have been more complete. I support publication of the study in the current form given that the authors elaborate on the caveats in the discussion as I pointed out above.

Thanks for the suggestions. We have further emphasized issues around m6A factor knockout in the discussion (p19). As an aside we tried very hard but failed to generate the homozygous Ythdc1 knockout, finally concluding that this must have a very severe cell lethal phenotype.

Reviewer #3

This manuscript by Nesterova et al. describes and compares the role of several factors in the X chromosome inactivation process. Previous work of the Brockdorff and other laboratories have implicated several chromatin remodelling factors to play a crucial role in this process, and in this study the individual role of Spen, Rbm15, WTAP, PRC1, PRC2 and many other factors have been investigated. For their studies they employ two inducible systems with a dox inducible Xist gene located on chromosome 3 in male cells, or an inducible endogenous Xist locus in female ES cells. Their studies indicate Spen is the central player in the XCI process acting through the A repeat in the 5' region of Xist. Other factors such as WTAP and LBR appear to play less important or redundant roles in the XCI process. Overall this is a very good and thorough study of importance for the XCI field and beyond. I have a few suggestions and questions to improve the manuscript.

Regarding the induction the authors should explain more thoroughly how the experiment is performed. Is dox induction started prior to differentiation and for how long? Also in comparison to previous studies of several laboratories using ES cells as model system silencing in this present study appears to happen very quickly.

Thanks for the suggestions. We have amended the methods section accordingly.

Part of the X chromosome appears to be non-informative. Why is this were these F1 or N1 129:cas ESCs?

A mitotic recombination event close to Xist that occurred during derivation of our engineered cell line resulted in the distal part of the X chr being non-informative for SNPs. This is described in results, p 6, end of paragraph 1.

In Suppl. Figure 2f the authors claim a correlation between silenced genes in Spen KO and delta-A cells. To me this correlation is not very clear at all despite a significance value for the hypergeometric test, why not use pearson or spearman here?

We didn't use Pearson/Spearman because the majority of points are for genes with no silencing and their values therefore represent noise. When we do apply these tests we indeed see only a weak correlation ($r=0.23$). We therefore used enriched analysis (Hypergeometric test) to calculate the significance whether less-sensitive genes in SpenKO are enriched among less-sensitive genes in XistDA.

Why is the data represented Xi vs Xi+Xa in Figure 1, whereas in the other Figures dox- vs dox + is shown. This needs explanation.

The reason for this was to illustrate the baseline (no dox) on the graphical representations. Thereafter we simply subtracted no dox values for comparison purposes.

The authors claim that 'silencing was abrogated to a greater degree after 6 days of Xist RNA induction in differentiating conditions (Fig. 6c-e)', however the relative ratio vs control remains the same, so what does this mean?

We performed ChrRNA-seq upon the 1 day of induction of Xist in WT and XistDelPID cells, as well as 6 days differentiation, then we calculated the allelic ratio (further Dox – NoDox, as mentioned above and see methods as well) for each X-linked gene in WT and XistDelPID cells. To estimate the silencing deficiency, we further calculate the difference between XistDelPID and WT cells by gene silencing degree. This difference enlarges at 6 days compared to 1 day Xist induction. See the right panel of Fig.6e. A rationale for this effect comes from a recent collaborative study in which we show that the PID region is required for recruitment of the late stage silencing factor Smchd1 (Jansz et al, Cell Reports, 2018), a finding that we have added to our discussion. See also comments to reviewer 1.

It is unfortunate that the Suz12 KO was only examined in the chr3 context, but I guess this is a consequence of how in real time the order of experiments was performed.

Agreed, although we are confident that the chr3 model is valid for determining the relative contribution of PRC1 and PRC2 in Xist-mediated silencing.

REVIEWERS' COMMENTS:

Reviewer #1 (Remarks to the Author):

Nesterova et al improved their revised manuscript by answering most of the reviewers' comments and discussing the different caveats.

I appreciate the effort made by the authors to include the mutant ESC controls. However, most of the RNA-FISH and IF experiments are still missing a proper quantification as well as the number of cells analysed (Supplemental Figures 2d, 3b, 4d, 6a and Figure 5d).

I am supporting publication of this manuscript once adequate quantifications will be added.

Reviewer #3 (Remarks to the Author):

The authors have satisfactorily addressed all my questions and concerns.

REVIEWERS' COMMENTS:

Reviewer #1 (Remarks to the Author):

Nesterova et al improved their revised manuscript by answering most of the reviewers' comments and discussing the different caveats.

I appreciate the effort made by the authors to include the mutant ESC controls. However, most of the RNA-FISH and IF experiments are still missing a proper quantification as well as the number of cells analysed (Supplemental Figures 2d, 3b, 4d, 6a and Figure 5d).

Quantification for all cell lines is added to the figure 5 and Supplementary figures 1-4 and 5.

Figure legends have been amended accordingly.

I am supporting publication of this manuscript once adequate quantifications will be added.

Reviewer #3 (Remarks to the Author):

The authors have satisfactorily addressed all my questions and concerns.